# Turning Tabular Foundation Models into Graph Foundation Models

## Abstract

While foundation models have revolutionized such fields as natural language processing and computer vision, their potential in graph machine learning remains largely unexplored. One of the key challenges in designing graph foundation models (GFMs) is handling diverse node features that can vary across different graph datasets. While many works on GFMs have focused exclusively on text-attributed graphs, the problem of handling arbitrary features of other types in GFMs has not been fully addressed. However, this problem is not unique to the graph domain, as it also arises in the field of machine learning for tabular data. In this work, motivated by the recent success of tabular foundation models (TFMs) like TabPFNv2 or LimiX, we propose G2T-FM, a simple framework for turning tabular foundation models into graph foundation models. Specifically, G2T-FM augments the original node features with neighborhood feature aggregation, adds structural embeddings, and then applies a TFM to the constructed node representations. Even in a fully in-context regime, our model achieves strong results, significantly outperforming publicly available GFMs and performing competitively with, and often better than, well-tuned GNNs trained from scratch. Moreover, after finetuning, G2T-FM surpasses well-tuned GNN baselines. In particular, when combined with LimiX, G2T-FM often outperforms the best GNN by a significant margin. In summary, our paper reveals the potential of a previously overlooked direction of utilizing tabular foundation models for graph machine learning tasks.[1]

## 1 Introduction

In recent years, foundation models have become a major breakthrough in deep learning. Foundation models are large machine learning models that are pretrained on diverse and extensive datasets. After this pretraining phase, they can be easily adapted to a variety of specific tasks with minimal additional training. Well-known examples include BERT (Devlin et al., 2019) and GPT (Brown et al., 2020) in natural language processing, as well as CLIP (Radford et al., 2021) in computer vision. The core principle behind foundation models is to learn general representations by leveraging large and varied data. These representations capture important patterns and semantics in the data, making the pretrained models highly transferable to different tasks. As a result, foundation models consistently achieve state-of-the-art results, while also improving efficiency and generalization. Furthermore, these models unify techniques across different fields, driving rapid progress and innovation in deep learning research and applications.

Despite their remarkable success in such areas as computer vision and natural language processing, the development of foundation models for graph data has been less advanced. The challenges of developing graph foundation models (GFMs) stem from the fact that graphs are not actually a single domain, but rather a way to represent data from different domains. These domains use graphs to represent very different structures, such as social networks, web networks, road networks, co-purchasing networks, molecules, connectomes, or even abstract objects and their relations. Thus, successful GFMs should be able to work with graphs from different domains representing very different objects with nodes and very different relations with edges, which is a rather formidable task that requires overcoming many serious challenges. Two key challenges faced by GFMs are the ability to transfer to new feature spaces and target spaces. Graphs from different domains often have

---

[1]Our source code is available at https://anonymous.4open.science/r/8456b37f41900b.

different node features and different targets, making it difficult to design GFMs that can work across various types of graphs. Some existing GFMs restrict themselves to text-attributed graphs (Wang et al., 2024b; He et al., 2025; Liu et al., 2024), which allows them to use pretrained text encoders. Another approach is to use simple dimensionality reduction methods like SVD and PCA (Xia & Huang, 2024; Zhao et al., 2024a; Wang et al., 2025a; Yu et al., 2025), which allow transforming all feature spaces to a space with a fixed predefined number of features. However, these approaches do not allow for fully and effectively leveraging arbitrary node features in graphs from new domains.

The challenges of transferring to new feature and target spaces are not, however, exclusive to graphs. Tabular data — one of the most widespread data modalities in machine learning — is similar to graph-structured data in that it does not constitute a single domain but is a way to represent data from different domains. Thus, tabular datasets come with different feature and target spaces, so tabular foundation models face similar issues to GFMs. While tabular foundation models are not as developed as foundation models for language or vision, they have seen increased interest recently (Van Breugel & Van Der Schaar, 2024), with the first successful approaches being proposed (Hollmann et al., 2023; 2025; Mueller et al., 2025; Ma et al., 2024; Qu et al., 2025). For instance, TabPFNv2 (Hollmann et al., 2025) demonstrates strong performance in both in-context and finetuning regimes, and it has recently gained significant attention from the community.

This parallel suggests that developers of GFMs can draw inspiration from tabular foundation models, as they have to deal with many of the same challenges. In this paper, we take a first step in this direction and show that tabular foundation models, such as TabPFNv2 (Hollmann et al., 2025) and LimiX (Zhang et al., 2025), can be effectively adapted to graph datasets. We introduce a simple framework named Graph-to-Table Foundation Model (G2T-FM), which transforms graph tasks into tabular ones and solves them with a tabular foundation model. More specifically, we augment the original features with neighborhood feature aggregations (Bazhenov et al., 2025), classical structure-based features (node degree, PageRank, and the eigenvectors of the graph Laplacian), and learnable structure-based encodings (Kanatsoulis et al., 2025). Then, we apply a tabular foundation model to the constructed node representations to get predictions.

Our empirical results indicate that this straightforward framework achieves strong results in a fully in-context regime, significantly outperforming existing publicly available GFMs and performing competitively with, and often better than, well-tuned GNNs trained from scratch. Moreover, after finetuning, G2T-FM surpasses well-tuned GNN baselines, with especially strong improvements obtained when G2T-FM uses LimiX as a tabular foundation model (see Table 2). These results highlight the potential of the proposed approach and the positive transfer brought by the usage of foundation models.

Our main contributions are as follows:

- We identify a promising and previously overlooked direction of applying tabular foundation models to graph machine learning.
- As a proof of concept, we introduce G2T-FM, a simple framework that uses a tabular foundation model (TabPFNv2 and LimiX in our experiments) as the backbone of a graph foundation model.
- We show that, despite its simplicity, G2T-FM is a strong baseline for GFMs, substantially outperforming the existing publicly available GFMs.
- We further demonstrate that finetuned G2T-FM surpasses traditional GNNs trained from scratch.

We hope that our study will stimulate further development of generalizable and robust graph foundation models and encourage further adoption of tabular foundation models for graph-structured data, as well as possibly for other data modalities.

## 2 BACKGROUND

### 2.1 GRAPH FOUNDATION MODELS FOR NODE CLASSIFICATION

Graph foundation models (GFMs) have recently gained significant attention in the field of graph machine learning. The main purpose of GFMs is to enable effective transfer of knowledge across different graph datasets. In other words, they aim to learn knowledge from a variety of graph tasks that can be successfully applied to other graphs.

In this work, we primarily focus on node-level tasks such as node classification and node regression. While many GFMs are limited to text-attributed graphs (TAGs) (Wang et al., 2024b; He et al., 2025; Liu et al., 2024), they are not truly general as graphs in many domains involve non-textual features. Therefore, we specifically discuss methods applicable to graphs with arbitrary numerical and categorical node features which frequently appear in various real-world industrial applications.

In the following sections, we review the key design choices and considerations in the development of GFMs. In particular, we focus on pretraining objectives and data, as well as how GFMs handle graph structure and node features. For further details, see the survey by Wang et al. (2025b).

**Pretraining objective**  Some graph foundation models use self-supervised learning (SSL) objectives to guide their pretraining process (Zhao et al., 2024a; Yu et al., 2025; Wang et al., 2025a), whereas others employ supervised learning strategies (Lachi et al., 2024; Finkelshtein et al., 2025). Notably, several works (Xia et al., 2024; Xia & Huang, 2024) reduce the node classification task to link prediction. Specifically, for each label in a downstream task, they create a virtual node that is connected to all training nodes of that class. Node classification thus becomes a task of predicting links to those virtual class nodes, for which the models are pretrained.

**Pretraining data**  Collecting a sufficiently diverse collection of datasets for pretraining GFMs remains a significant challenge. To address this, some studies (Fey et al., 2025; Lachi et al., 2024; Xia et al., 2024) incorporate synthetic data — either together with real-world data or as an alternative. This includes the use of simple random graph models like stochastic block models (Lachi et al., 2024), as well as synthetic graphs generated by large language models (Xia et al., 2024). However, the majority of graph foundation models rely primarily on real-world datasets for pretraining. The number of datasets used for this purpose varies from as few as one graph (Finkelshtein et al., 2025), to more moderate collections of 2–10 datasets (Zhao et al., 2024a; Wang et al., 2025a), and up to several dozens in some studies (Xia & Huang, 2024; Lachi et al., 2024).

**Handling features**  One of the key challenges for graph foundation models is handling heterogeneous features that can vary significantly across different datasets. Some approaches address this by focusing exclusively on text-attributed graphs (TAGs), sometimes additionally converting non-textual features to text, and then applying a text encoder (Wang et al., 2024b; He et al., 2025; Liu et al., 2024). Methods aiming to deal with arbitrary features often rely on simple dimensionality reduction techniques such as SVD or PCA to obtain feature embeddings (Xia & Huang, 2024; Zhao et al., 2024a; Wang et al., 2025a; Yu et al., 2025). There are alternatives, such as learning dimension encoding modules that produce feature transformations (Zhao et al., 2024b), learning graph patches (Sun et al., 2025) or replacing node attribute values with their statistical dependencies (Shen et al., 2025), but these appear less common. We also highlight Finkelshtein et al. (2025), which constructs separate embeddings for each (node, feature) pair, enabling a more fine-grained representation of feature information.

**Handling structure**  Handling the structure is more straightforward, as graph neural networks are particularly well-suited for this task, and they are inherently capable of processing arbitrary graph structures. Consequently, many GFMs simply adopt GNNs as their backbone to handle graph structure (Zhao et al., 2024a; Finkelshtein et al., 2025; Yu et al., 2025; Wang et al., 2025a). In addition to GNN-based approaches, some methods use matrix decomposition techniques such as SVD applied to graph-derived matrices (for example, the normalized adjacency matrix or the sum of its powers), to encode structural information (Xia et al., 2024). However, while GNNs can in principle operate on any graph, their performance may still be limited due to varying graph structures. To address this, some works implement additional mechanisms specifically designed to handle structural differences (Yu et al., 2025; Wang et al., 2025a).

## 2.2 LIMITATIONS OF EXISTING GFMS

**Focus on text-attributed graphs**  Many existing graph foundation models are specifically designed for text-attributed graphs, where nodes or edges have associated textual information (Wang et al., 2024b; He et al., 2025; Liu et al., 2024). These models typically leverage large language models or other text encoders to process textual attributes, integrating natural language representations with graph structures. While this approach can be effective for certain domains such as academic networks or knowledge graphs, it limits the applicability of GFMs across a broader range of graphs

where such text attributes are not available. For instance, for graphs representing transportation networks, biological networks, or transaction networks (commonly used for fraud detection tasks), which often come with rich numerical and categorical features, the reliance exclusively on textual information restricts the model's usability and effectiveness. As a result, many current GFMs may not generalize well to graphs with non-textual attributes, hindering their adoption across diverse real-world scenarios.

**Limited support for regression tasks** Most publicly available GFMs are designed and evaluated on classification tasks, where the goal is to predict categorical labels for nodes, edges, or entire graphs. To date, no popular GFMs, aside from TS-GNN (Finkelshtein et al., 2025), support regression tasks, where the output is a continuous value rather than a class label. This is a substantial limitation because many important graph-based applications require regression instead of classification. The lack of support for regression tasks reduces the practical applicability of current GFMs and highlights an important area for future research.

**Misleading use of the "zero-shot" term** Some recent studies on graph foundation models have described their methods as operating in a "zero-shot" setting (Xia & Huang, 2024; Xia et al., 2024). Typically, these approaches introduce virtual nodes that represent target classes and connect them to the corresponding real nodes with known class labels. Then, the node classification problem reduces to predicting links between the test nodes and the appropriate virtual nodes. This process makes it possible to perform evaluation on unseen graphs without additional finetuning. While inventive and interesting, this technique does not truly realize zero-shot learning. Strictly speaking, zero-shot learning means that no labeled examples of the target classes are available during evaluation. However, the described method requires labeled nodes to be connected to virtual class nodes for effective link prediction. Therefore, the correct term for this setup should be "in-context learning", since evaluation does not involve further finetuning but still depends on access to labeled training samples. This inconsistency in terminology may lead to misleading comparisons with baseline approaches. For instance, the aforementioned studies (Xia & Huang, 2024; Xia et al., 2024) compare their "zero-shot" performance against the one-shot and five-shot results of other baselines, yet they do not clearly report the number of training samples used in "zero-shot" evaluation of the proposed method, which makes the comparison harder to interpret.

### 2.3 TABULAR FOUNDATION MODELS

The field of tabular foundation models (TFMs) was pioneered by the TabPFN model (Hollmann et al., 2023) that was designed to address any tabular problem off-the-shelf. TabPFN employs a transformer-like architecture and works in the in-context learning regime, with the entire downstream training set serving as the prompt. The pretraining of TabPFN was performed on a large number of synthetic datasets designed to mimic typical tabular tasks. The more recent model, TabPFNv2 (Hollmann et al., 2025), employs a more powerful backbone architecture, pretraining on a broader spectrum of synthetic datasets, and advanced techniques of data preprocessing. Nowadays, new TFMs are emerging regularly (Mueller et al., 2025; Ma et al., 2024; Qu et al., 2025; Zhang & Robinson, 2025; Zhang et al., 2025), and their success is exploited beyond the domain of pure tabular tasks, e.g., for time series forecasting (Hoo et al., 2025). In our work, we demonstrate that TFMs can also serve as a core building block for graph foundation models.

### 3 GRAPH-TO-TABLE FOUNDATION MODEL

As discussed above, one of the key challenges for graph foundation models is processing node features that can vary widely across different graphs and domains. To address this, previous approaches primarily rely on one of the following strategies. The first is to apply dimensionality reduction techniques, such as principal component analysis or singular value decomposition (Xia & Huang, 2024; Zhao et al., 2024a; Wang et al., 2025a; Yu et al., 2025). However, this approach can result in the loss of information, and, although it ensures that all feature vectors share the same dimension after reduction, it remains unclear whether these reduced features are transferable across different graphs. The second strategy employs text encoders to process node or edge features (Wang et al., 2024b; He et al., 2025; Liu et al., 2024). This method is highly effective for text-attributed graphs (TAGs), where the features are naturally represented as text. Nevertheless, many real-world graphs do not include solely text features (Ivanov & Prokhorenkova, 2021; Chen et al., 2022; Robinson et al., 2024;

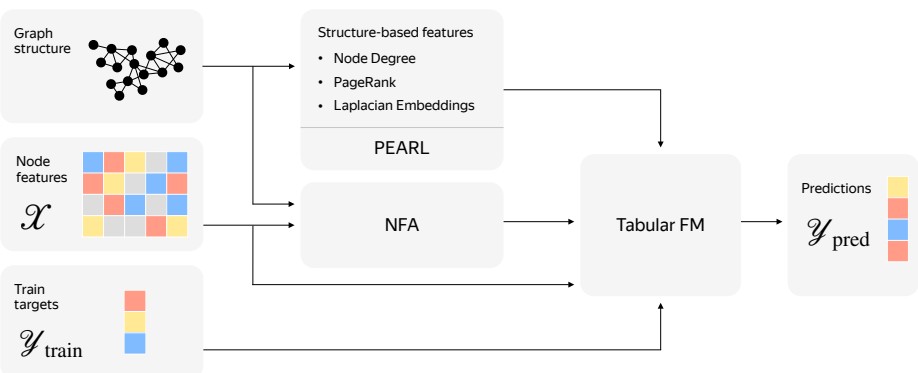

Figure 1: Overview of the proposed G2T-FM framework.

Wang et al., 2024a; Bazhenov et al., 2025). Using text encoders for non-text features can therefore be highly suboptimal, as it does not leverage the nature or structure of these features. Overall, neither of these strategies fully addresses the problem of handling diverse and heterogeneous features.

Similarly, the problem of adapting to different target spaces has also not been fully solved. Existing approaches, such as converting node classification to link prediction or using textual descriptions of node classes, only work with node classification tasks and not with node regression tasks, which frequently appear in real-world applications of graph machine learning.

However, these challenges are not unique to graph machine learning, and they also arise in tabular machine learning. Recent advances in tabular deep learning, such as the development of foundation models like TabPFNv2 (Hollmann et al., 2025), offer promising solutions for handling diverse feature and target spaces.

We argue that tabular foundation models can be used to create better graph foundation models, and, in particular, they can help handle different feature and target spaces. As a proof of concept, we introduce Graph-to-Table Foundation Model (G2T-FM) framework, which addresses the challenge of learning on graphs with diverse and heterogeneous node features and different targets. Figure 1 provides an overview of our method.

To process heterogeneous node features, we employ a tabular foundation model like TabPFNv2 (Hollmann et al., 2025) or LimiX (Zhang et al., 2025). TFMs are designed for tabular data only, so to make them applicable to graph-structured data, we introduce a graph-based preprocessing step that encodes graph information into the node features. Our goal is to capture the information about different aspects of the graph structure as well as the interplay between the graph structure and the node features. Hence, the new augmented feature vector consists of the original node features and the following graph-based components.

**Neighborhood feature aggregation (NFA)** Following Bazhenov et al. (2025), for each node, we compute aggregated feature statistics over its one-hop neighbors. Namely, for each numerical feature, we compute its mean, maximum, and minimum values over the node's neighbors. For each categorical feature, we first apply one-hot encoding and then compute the mean of the obtained binary features. The computation of NFA uses both the graph structure and node features. This component provides information about the features in the local neighborhood of the node, which is a valuable signal for many graph-related tasks.

**Classic structure-based features (SF)** We also include basic node structural characteristics — node degree, PageRank score, and Laplacian eigenvectors. The first two are classic node centrality measures that indicate how "important" a particular node is; however, they do so in different ways. The node degree captures strictly local information, while PageRank also captures global information. Then, we compute the first $K$ eigenvectors of the graph Laplacian and consider the corresponding $K$-dimensional embeddings as additional feature vectors. Laplacian embeddings en-

code a node's position within the graph relative to other nodes. Thus, such node representations provide valuable information that supplements the centrality measures.

**Learnable structure-based encodings (PEARL)** These encodings have been proposed in Kanatsoulis et al. (2025). The basic idea of PEARL is to generate a random value for each node and then apply a GNN using these values as node features. Such random initialization increases the expressive power of GNNs by breaking the structural symmetries. This procedure is repeated $M$ times (each with new random node features) and the resulting node embeddings are averaged so that node permutation equivariance still holds in the limit. Kanatsoulis et al. (2025) propose using this as a learnable module, so that the encodings are trained to improve downstream performance. However, we noticed that PEARL encodings are also useful without training, i.e., when we use a randomly initialized GNN to obtain node embeddings. Thus, this module can produce both non-learnable and learnable representations, where non-learnable representations do not involve parameter optimization (training).

These new features allow us to provide the tabular foundation model with information about many properties of the graph underlying the given dataset. We concatenate them with the original node attributes and use the resulting augmented feature vector as an input to TFM. The resulting model can be applied in a fully in-context regime and, as we show below, it performs on par with GNNs that are well-tuned for a single dataset. In the finetuning regime, we jointly tune PEARL and TFM.

As a final remark, we discuss the symmetries of G2T-FM. Following Finkelshtein et al. (2025), a graph foundation model should satisfy three natural inductive biases: (i) feature permutation invariance; (ii) label permutation equivariance; and (iii) node permutation equivariance. In Appendix D, we show that G2T-TabPFNv2 satisfies these symmetries in distribution: its outputs depend on internal randomness, but the distribution of the outputs remains invariant or equivariant under the corresponding permutations.

## 4 Experimental Setup

### 4.1 Datasets

In our experiments, we use TabPFNv2 as one of the backbones, which imposes specific constraints that direct our dataset selection. In particular, TabPFNv2 is suitable for both regression tasks and multi-class classification tasks, but the latter is restricted to at most 10 classes. Additionally, it is designed for small-to-medium-scale datasets, requiring no more than roughly 10,000 training samples. As such, we select datasets that fit within these boundaries. We note that future developments in tabular foundation models may relax these constraints and enable experiments with a broader range of graph benchmarks.

To comprehensively assess the capabilities of our method, we construct two collections of datasets. The first focuses on graphs with non-textual node features (our primary setting), while the second contains well-known graph benchmarks with text-based node features. Across these datasets, our selection covers both regression and classification tasks, and includes graphs with both homophilious and non-homophilous structure,[2] as summarized in Table 1.

**Datasets with non-textual features** This collection comprises eight datasets from the Graph-Land benchmark (Bazhenov et al., 2025), all featuring diverse tabular node features.[3] The datasets include: social networks `artnet-exp`, `artnet-views`, and `twitch-views`; a network of workers from a crowdsourcing platform `tolokers-2`; a network of users of a review service `city-reviews`; a co-purchasing network `hm-prices`; a network of devices `avazu-ctr`; and a road network `city-roads-M`.

**Datasets with text-based features** This collection includes five classical graph datasets where node features are derived from textual descriptions: a network of users of a question-answering website `questions` (Platonov et al., 2023b); a citation network `pubmed` (Yang et al., 2016); a social network `facebook` (Rozemberczki & Sarkar, 2020); a co-purchasing

---

[2]A graph is called homophilous if its edges tend to connect nodes with similar labels, see Newman (2003); Platonov et al. (2023a); Mironov & Prokhorenkova (2024) for details.

[3]By tabular features, we mean a mixture of numerical and categorical features with different distributions.

Table 1: The key statistics of the considered graph datasets.

| name | # nodes | # edges | # features | mean degree | # classes | homophily | feature type |
|------|---------|---------|-----------|-------------|-----------|-----------|--------------|
| tolokers-2 | 11,758 | 519,000 | 16 | 88.3 | 2 | no | tabular |
| city-reviews | 148,801 | 1,165,415 | 37 | 15.7 | 2 | yes | tabular |
| artnet-exp | 50,405 | 280,348 | 75 | 11.1 | 2 | no | tabular |
| hm-prices | 46,563 | 10,730,995 | 41 | 460.9 | N/A | no | tabular |
| avazu-ctr | 76,269 | 10,984,077 | 260 | 288.0 | N/A | no | tabular |
| city-roads-M | 57,073 | 107,104 | 26 | 3.8 | N/A | yes | tabular |
| twitch-views | 168,114 | 6,797,557 | 4 | 80.9 | N/A | no | tabular |
| artnet-views | 50,405 | 280,348 | 50 | 11.1 | N/A | no | tabular |
| pubmed | 19,717 | 44,324 | 500 | 4.5 | 3 | yes | text-based |
| facebook | 22,470 | 170,823 | 128 | 15.2 | 4 | yes | text-based |
| amazon-ratings | 24,492 | 93,050 | 300 | 7.6 | 5 | no | text-based |
| questions | 48,921 | 153,540 | 301 | 6.3 | 2 | no | text-based |
| wiki-cs | 11,701 | 215,603 | 300 | 36.9 | 10 | yes | text-based |

network `amazon-ratings` (Platonov et al., 2023b); and a network of Wikipedia pages `wiki-cs` (Mernyei & Cangea, 2020).

While specialized graph foundation models may yield better results for text-attributed graphs, including these datasets allows us to test the generalization of our approach. Also, we explicitly exclude datasets with bag-of-words (BoW) node features, as BoW is less relevant to modern text processing, while introducing additional challenges like high-dimensionality and sparsity.

For all datasets, we employ a standardized data splitting protocol, allocating 10% of the nodes to training, 10% to validation, and the remaining 80% to testing. For the GraphLand datasets, we use the official `RL` (random low) splits. For the remaining datasets, we employ the random stratified splitting, which ensures consistent class distributions across the splits. All the experiments are run in a transductive setting, which is standard for node property prediction in the graph domain. For binary classification tasks, we report average precision. For multiclass classification tasks, we report accuracy. For regression tasks, we report $R^2$. For all metrics, higher is better.

## 4.2 METHODS

In our experiments with G2T-FM, we adopt TabPFNv2 (Hollmann et al., 2025) and LimiX (Zhang et al., 2025) as backbone models. We refer to these models as G2T-TabPFNv2 and G2T-LimiX respectively. For comparison, we also evaluate the following baseline methods.

First, we include traditional supervised baselines trained from scratch for each dataset. These include four classic GNNs: GCN (Kipf & Welling, 2017), GraphSAGE (Hamilton et al., 2017), GAT (Veličković et al., 2018), and neighborhood-attention Graph Transformer (GT) (Shi et al., 2021).[4] For all GNNs, we use the modifications from Platonov et al. (2023b) that augment GNNs with residual connections (He et al., 2016), layer normalization (Ba et al., 2016), and MLP blocks, which often significantly improve their performance. For the GraphLand datasets with tabular features, we also use LightGBM+NFA as a strong baseline — which is a popular implementation of gradient-boosted decision trees (Ke et al., 2017) with input features augmented with graph neighborhood information via NFA (Bazhenov et al., 2025). The implementation of these models closely resembles that from GraphLand (Bazhenov et al., 2025) in terms of both model architecture and hyperparameter tuning, see Appendix B for more details.

Second, we employ several publicly available graph foundation models. Despite significant interest in developing GFMs recently, most of the research is focused exclusively on text-attributed graphs (as discussed above), so we were able to find only a few openly available models that support node property prediction in graphs with arbitrary node feature spaces. Specifically, we employ AnyGraph (Xia & Huang, 2024), OpenGraph (Xia et al., 2024), and TS-GNN (Finkelshtein et al., 2025), which are used in the in-context learning (ICL) regime, as well as GCOPE (Zhao et al., 2024a), which is used in the finetuning (FT) regime. For all these methods, we use the original

---

[4]Neighborhood-attention GT uses only local attention to a node's neighbors, in contrast to graph transformers with global all-to-all attention.

Table 2: Evaluation results on datasets with tabular features (datasets from the GraphLand benchmark under the `RL` (random low) data split). For each column, we highlight first, second, and third best results with a color.

| | tolokers-2 | city-reviews | artnet-exp | hm-prices | avazu-ctr | city-roads-M | twitch-views | artnet-views | **AR** |
|---|---|---|---|---|---|---|---|---|---|
| LightGBM+NFA | $56.34 \pm 0.06$ | $78.53 \pm 0.01$ | $46.13 \pm 0.03$ | $70.84 \pm 0.04$ | $31.71 \pm 0.01$ | $61.18 \pm 0.03$ | $60.14 \pm 0.01$ | $56.10 \pm 0.02$ | **5.38** |
| GCN | $56.27 \pm 0.29$ | $77.81 \pm 0.14$ | $44.86 \pm 0.34$ | $68.02 \pm 0.40$ | $32.00 \pm 0.15$ | $58.82 \pm 0.24$ | $75.51 \pm 0.05$ | $56.03 \pm 0.24$ | **6.25** |
| GraphSAGE | $54.43 \pm 0.32$ | $78.17 \pm 0.09$ | $45.14 \pm 0.34$ | $70.00 \pm 0.70$ | $31.44 \pm 0.15$ | $59.44 \pm 0.26$ | $66.29 \pm 0.31$ | $49.32 \pm 0.86$ | **7.12** |
| GAT | $57.41 \pm 0.80$ | $77.74 \pm 0.20$ | $45.06 \pm 0.49$ | $72.07 \pm 1.16$ | $32.63 \pm 0.16$ | $59.86 \pm 0.19$ | $72.89 \pm 0.25$ | $53.60 \pm 0.23$ | **5.12** |
| GT | $56.98 \pm 0.53$ | $77.34 \pm 0.20$ | $46.41 \pm 0.68$ | $69.44 \pm 0.89$ | $31.11 \pm 0.47$ | $59.55 \pm 0.27$ | $72.13 \pm 0.13$ | $53.37 \pm 0.43$ | **6.62** |
| OpenGraph (ICL) | $40.38 \pm 1.13$ | $59.09 \pm 0.72$ | $15.16 \pm 0.83$ | N/A | N/A | N/A | N/A | N/A | **11.00** |
| AnyGraph (ICL) | $28.75 \pm 3.56$ | $63.71 \pm 1.45$ | $12.84 \pm 0.93$ | N/A | N/A | N/A | N/A | N/A | **12.33** |
| TS-GNN (ICL) | $38.54 \pm 0.94$ | $43.46 \pm 5.17$ | $20.44 \pm 1.05$ | N/A | N/A | N/A | N/A | N/A | **11.33** |
| GCOPE (FT) | $28.81 \pm 1.28$ | $67.16 \pm 0.98$ | $14.92 \pm 1.56$ | N/A | N/A | N/A | N/A | N/A | **11.33** |
| G2T-TabPFNv2 (ICL) | $60.42 \pm 0.27$ | $77.46 \pm 0.10$ | $45.84 \pm 0.03$ | $66.68 \pm 0.09$ | $26.38 \pm 0.07$ | $60.47 \pm 0.04$ | $70.00 \pm 0.06$ | $58.75 \pm 0.15$ | **6.38** |
| G2T-TabPFNv2 (FT) | $57.65 \pm 1.92$ | $79.12 \pm 0.21$ | $47.31 \pm 0.59$ | $71.05 \pm 0.91$ | $28.52 \pm 0.43$ | $63.08 \pm 0.28$ | $74.06 \pm 0.16$ | $60.29 \pm 0.13$ | **3.62** |
| G2T-LimiX (ICL) | $61.48 \pm 0.30$ | $77.72 \pm 0.54$ | $48.43 \pm 0.18$ | $74.96 \pm 0.06$ | $32.39 \pm 0.14$ | $64.53 \pm 0.07$ | $71.08 \pm 0.07$ | $60.95 \pm 0.10$ | **3.12** |
| G2T-LimiX (FT) | $61.17 \pm 0.49$ | $80.13 \pm 0.05$ | $49.88 \pm 0.13$ | $76.32 \pm 0.17$ | $33.94 \pm 0.34$ | $65.87 \pm 0.10$ | $73.16 \pm 0.40$ | $62.12 \pm 0.10$ | **1.38** |

implementations provided by the authors. Further, we were only able to evaluate these methods on datasets with node classification tasks, as none of them support node regression.[5]

For all the methods, we run experiments 10 times (5 times for GFMs from prior literature) and report the mean and standard deviation of the model performance, since the results are affected by stochasticity during model training and inference: some of the considered baselines have stochastic inference by design, while others require training or finetuning with different random states.

## 5 EXPERIMENTAL RESULTS

Table 2 contains the evaluation results on graph datasets with tabular features, and Table 3 contains the additional results on datasets with text-derived features. In addition to the results on individual datasets, we also report the average ranks (AR). Below, we summarize and discuss our key observations.

> **Observation 1** *In our evaluation, the existing publicly available graph foundation models perform substantially worse than well-tuned traditional GNNs trained from scratch.*

This observation holds true across both collections of datasets we evaluated. The sole exception we identified is the performance of TS-GNN on the `amazon-ratings` dataset, where it surpassed the GNNs trained from scratch. In all other cases, the performance of the existing GFMs was significantly lower than that of our GNN baselines.

While many GFM publications report outperforming traditional GNNs, our findings suggest otherwise. We hypothesize that this discrepancy stems from several key differences in the evaluation protocol. First, unlike some GFM studies that focus on few-shot benchmarks, we use different datasets and larger training splits (10%/10%/80%) that allow GNNs to be trained effectively from scratch. Second, we ensure our GNN baselines are highly competitive by performing a thorough hyperparameter optimization and using the GNN architectures from Platonov et al. (2023b) that include established performance-enhancing features like residual connections and normalization (Luo et al., 2024; 2025), which are often absent in simpler baselines.

> **Observation 2** *On datasets with tabular features, G2T-FM evaluated in the in-context learning mode performs competitively with, and often better than, traditional GNNs trained from scratch.*

In particular, in terms of the average rank on datasets with tabular features, G2T-LimiX outperforms all baselines trained from scratch, while G2T-TabPFNv2 performs on par with them. Results of G2T-FM on individual datasets are also strong. For example, on `tolokers-2`, `artnet-exp`, `hm-prices`, `city-roads-M`, and `artnet-views`, G2T-LimiX surpasses all traditional baselines, often by more than two percentage points.

---

[5] While TS-GNN in theory supports node regression, its current official implementation, to the best of our knowledge, does not allow running experiments for regression tasks.

Table 3: Evaluation results on datasets with text-based features. For each column, we highlight first, second, and third best results with a color.

|  | pubmed | facebook | amazon-r. | questions | wiki-cs | AR |
|---|---|---|---|---|---|---|
| GCN | 85.46 ± 0.18 | 91.26 ± 0.19 | 41.43 ± 0.46 | 15.42 ± 0.63 | 81.74 ± 0.20 | 5.60 |
| GraphSAGE | 86.04 ± 0.26 | 91.12 ± 0.21 | 40.07 ± 0.50 | 16.55 ± 0.61 | 81.50 ± 0.26 | 6.00 |
| GAT | 84.81 ± 0.22 | 92.61 ± 0.20 | 40.67 ± 0.53 | 16.75 ± 0.63 | 82.25 ± 0.26 | 4.20 |
| GT | 84.95 ± 0.18 | 91.71 ± 0.21 | 41.56 ± 0.38 | 14.03 ± 0.86 | 82.54 ± 0.20 | 5.00 |
| OpenGraph (ICL) | 70.30 ± 2.67 | 75.27 ± 5.05 | 29.36 ± 1.24 | 3.77 ± 0.65 | 75.66 ± 0.39 | 10.80 |
| AnyGraph (ICL) | 65.31 ± 6.26 | 61.17 ± 8.64 | 33.49 ± 3.44 | 4.27 ± 0.66 | 65.17 ± 2.51 | 11.00 |
| TS-GNN (ICL) | 64.41 ± 5.11 | 77.87 ± 2.73 | 43.00 ± 0.13 | 5.00 ± 0.48 | 46.25 ± 9.77 | 9.60 |
| GCOPE (FT) | 79.35 ± 0.70 | 85.08 ± 0.17 | 39.90 ± 0.43 | 6.59 ± 0.43 | 59.13 ± 1.20 | 9.60 |
| G2T-TabPFNv2 (ICL) | 88.80 ± 0.25 | 90.56 ± 0.12 | 40.63 ± 0.19 | 16.49 ± 0.16 | 76.61 ± 0.57 | 6.60 |
| G2T-TabPFNv2 (FT) | 90.46 ± 0.11 | 91.73 ± 0.28 | 44.71 ± 0.32 | 19.07 ± 0.53 | 79.70 ± 0.31 | 3.00 |
| G2T-LimiX (ICL) | 88.96 ± 0.18 | 91.29 ± 0.14 | 44.10 ± 0.16 | 15.31 ± 0.77 | 79.99 ± 0.28 | 4.80 |
| G2T-LimiX (FT) | 89.91 ± 0.48 | 92.16 ± 0.18 | 45.67 ± 0.35 | 20.19 ± 0.30 | 82.24 ± 0.31 | 1.80 |

At the same time, in-context performance of both G2T-TabPFNv2 and G2T-LimiX on datasets with text-based features is less impressive. In particular, G2T-TabPFNv2 is outperformed by all traditional GNNs. We attribute this to TFMs being trained on tabular data: non-tabular features appear to be out-of-domain, and good performance therefore requires adaptation via finetuning.

> **Observation 3** *After finetuning, G2T-FM outperforms on average all traditional baselines trained from scratch on both collections of datasets with G2T-LimiX achieving especially strong improvements.*

On datasets with tabular features, G2T-LimiX achieves not only the best average rank, but also the best results on six out of eight datasets, often yielding notable improvements of more than two percentage points compared to the best GNN result.

On datasets with text-based features, finetuned G2T-FM is also strong. In particular, G2T-LimiX achieves the best average rank and often brings noticeable improvements over traditional GNNs. We note, however, that specialized models that process raw text more directly may achieve higher accuracy on text-attributed graphs. Though, comparing against such models is outside the scope of this work.

Finally, we refer to Appendix C for the ablation analysis. Our results show that the gains of G2T-FM come from the synergy between the TFM backbone and our graph-to-table components.

## 6 CONCLUSION

In this work, we have shown that tabular foundation models can be successfully employed for solving graph problems since they are able to process heterogeneous feature and target spaces. To show this, we proposed a simple G2T-FM framework, which converts a graph task into a tabular task by augmenting the initial features with graph information, and then applies a tabular foundation model. Our empirical results show the strong performance of G2T-FM, both in the in-context and finetuning regimes. In particular, G2T-LimiX outperforms all well-tuned GNN baselines in terms of the average rank even in the in-context learning regime. After finetuning, G2T-LimiX achieves the best results on most of the datasets, often outperforming the best GNN by a significant margin. Our approach is simple, but it provides significant advantages over prior attempts at developing GFMs and shows the potential of creating truly generalizable GFMs that can achieve strong results across diverse tasks and real-world applications of graph machine learning, regardless of the inherent feature and target spaces. Importantly, G2T-FM can be combined with any tabular foundation model. Thus, future advancements in TFMs can be easily transferred to the graph ML domain.

Despite its strong performance, G2T-FM framework is only a first step towards utilizing models and ideas from the tabular domain for developing truly generalizable GFMs. Hence, our work has several noteworthy limitations that suggest future research directions; we discuss those in Appendix A.

## REPRODUCIBILITY STATEMENT

We provide the code and instructions for reproducing all the experimental results in the following anonymous repository: https://anonymous.4open.science/r/8456b37f41900b. We also discuss G2T-FM implementation in Section 3 and Appendix B, while details on evaluation are provided in Section 4 and Appendix B.

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

## A    LIMITATIONS AND FUTURE WORK

Despite its strong performance, the proposed G2T-FM framework is only a first step towards utilizing models and ideas from the tabular domain for developing truly generalizable GFMs. Hence, our work has several noteworthy limitations that suggest future research directions.

First, the present version of G2T-FM uses only basic methods for processing graph structures. Future research can bring more graph-specific components into the framework, such as more complex aggregation mechanisms (including learnable and multi-hop aggregations) and cross-graph pretraining. We believe these extensions could enable the model to better capture graph-specific information and transfer knowledge across different graphs.

Second, G2T-FM inherits several restrictions from its TFM backbone. In particular, when endowed with TabPFNv2, G2T cannot handle classification datasets with more than 10 classes, and its training set size is limited to at most 10,000 samples. These limitations currently make it difficult to apply G2T-FM to large-scale datasets. However, future research on tabular foundation models can alleviate these limitations and the new TFMs can be directly used within the G2T-FM framework.

Finally, the scope of our work is limited to node-level prediction tasks[6] and our method lacks graph-specific pretraining, which could be cited as reasons against classifying G2T-FM as a graph foundation model. Nevertheless, we position our work within the research area of graph foundation models due to the following reasons. First, G2T-FM can be applied to arbitrary node-level problems in an in-context learning setting, which is a property inherent to foundation models. Second, since GFMs represent an active area of research, the community currently lacks a strict and clear definition of a GFM. For example, some works argue that a GFM should be able to handle node/link/graph-level tasks at the same time, while others propose task-specific (e.g., node-level only) GFMs (Finkelshtein et al., 2025; Lachi et al., 2024; Zhao et al., 2024a; Wang et al., 2025b). Finally, our work aims to contribute to the GFM field by leveraging the recent advancements in TFM research, and we hope that our approach will inspire further innovations in GFM design and encourage researchers to explore similar ideas. Thus, we believe that positioning our work within the GFM context is both appropriate and beneficial for advancing the field.

## B    IMPLEMENTATION DETAILS

In this section, we describe our main implementation choices. Additional information and the full code are available in our repository.

### B.1    G2T-FM

**Finetuning**    For the finetuning experiments, we follow the procedure outlined by Rubachev et al. (2025). Rather than using parameter-efficient finetuning, we opt for full model finetuning, as previous work indicates this yields better performance. We search for the optimal learning rate over the logarithmic grid of 10 values, ranging from $5 \times 10^{-6}$ to $5 \times 10^{-4}$.

**PCA**    On certain datasets, specifically, `city-reviews` and `avazu-ctr`, applying G2T-FM directly on the original features results in out-of-memory errors. To address this, we apply principal component analysis (PCA) to reduce the feature dimensionality. PCA is performed separately on the original features and the neighborhood feature aggregations.

**PEARL**    For the GNN backbone within PEARL, our implementation is based on the GraphLand implementation (Bazhenov et al., 2025). However, we remove layer normalization and residual connections, based on preliminary experiments that showed improved results without these components.

For the in-context learning experiments, we utilize a randomly initialized PEARL model whose weights are shared across all datasets. Interestingly, even without explicit training, this untrained PEARL model still produces useful representations for some datasets, as demonstrated in our abla-

---

[6]Potentially, G2T-FM can be applied to link-level tasks, but its performance on such problems requires further analysis. Regarding graph-level tasks, how to adapt G2T-FM is less straightforward.

Table 4: Ablation of the components of G2T-FM. SF stands for Structure-based Features, which include degree, PageRank, and Laplacian eigenvectors.

| | tolokers-2 | city-reviews | artnet-exp | hm-prices | avazu-ctr | city-roads-M | twitch-views | artnet-views | **AR** |
|---|---|---|---|---|---|---|---|---|---|
| G2T-TabPFNv2 (ICL) | $60.42 \pm 0.27$ | $77.46 \pm 0.10$ | $45.84 \pm 0.03$ | $66.68 \pm 0.09$ | $26.38 \pm 0.07$ | $60.47 \pm 0.04$ | $70.00 \pm 0.06$ | $58.75 \pm 0.15$ | **5.88** |
| w/o NFA (ICL) | $60.28 \pm 0.34$ | $76.98 \pm 0.08$ | $43.57 \pm 0.13$ | $62.90 \pm 0.09$ | $25.54 \pm 0.33$ | $58.68 \pm 0.23$ | $69.61 \pm 0.05$ | $58.86 \pm 0.22$ | **7.62** |
| w/o SF & PEARL (ICL) | $56.17 \pm 0.12$ | $76.87 \pm 0.02$ | $45.90 \pm 0.01$ | $67.22 \pm 0.03$ | $26.78 \pm 0.02$ | $60.08 \pm 0.01$ | $58.94 \pm 0.06$ | $55.30 \pm 0.02$ | **7.88** |
| w/o SF (ICL) | $57.48 \pm 0.18$ | $77.07 \pm 0.02$ | $45.92 \pm 0.02$ | $67.11 \pm 0.03$ | $26.73 \pm 0.01$ | $60.01 \pm 0.07$ | $67.80 \pm 0.02$ | $56.23 \pm 0.03$ | **6.88** |
| w/o PEARL (ICL) | $60.56 \pm 0.32$ | $77.47 \pm 0.10$ | $45.84 \pm 0.03$ | $66.81 \pm 0.06$ | $26.33 \pm 0.09$ | $60.53 \pm 0.03$ | $65.30 \pm 0.07$ | $58.36 \pm 0.24$ | **6.12** |
| G2T-TabPFNv2 (FT) | $57.65 \pm 1.92$ | $79.12 \pm 0.21$ | $47.31 \pm 0.59$ | $71.05 \pm 0.91$ | $28.52 \pm 0.43$ | $63.08 \pm 0.28$ | $74.06 \pm 0.16$ | $60.29 \pm 0.13$ | **2.25** |
| w/o NFA (FT) | $59.75 \pm 0.76$ | $78.61 \pm 0.21$ | $45.10 \pm 0.35$ | $67.10 \pm 0.74$ | $25.98 \pm 0.99$ | $61.40 \pm 0.44$ | $73.28 \pm 0.15$ | $59.93 \pm 0.18$ | **5.38** |
| w/o SF & PEARL (FT) | $57.19 \pm 1.15$ | $78.68 \pm 0.18$ | $47.05 \pm 0.39$ | $71.19 \pm 0.61$ | $27.99 \pm 0.36$ | $62.67 \pm 0.17$ | $60.98 \pm 0.14$ | $57.02 \pm 0.45$ | **5.25** |
| w/o SF (FT) | $57.44 \pm 0.48$ | $78.61 \pm 0.13$ | $47.17 \pm 0.40$ | $71.77 \pm 0.53$ | $28.31 \pm 0.58$ | $59.72 \pm 0.70$ | $73.10 \pm 0.21$ | $58.29 \pm 0.10$ | **4.62** |
| w/o PEARL (FT) | $60.18 \pm 0.49$ | $79.18 \pm 0.21$ | $47.57 \pm 0.43$ | $70.64 \pm 0.83$ | $28.19 \pm 0.31$ | $63.28 \pm 0.25$ | $66.90 \pm 0.19$ | $60.26 \pm 0.12$ | **2.88** |

tion studies (Appendix C). For the finetuning experiments, we jointly finetune PEARL and the TFM backbone.

**Label shuffling**    To ensure that our framework is equivariant to permutations of class labels in multiclass classification tasks, we employ a label shuffling procedure. During each forward pass, class numerical labels are randomly shuffled, so that the average predictions remain independent of the original numerical label assignment.

### B.2   GNNs AND LIGHTGBM

**GNNs**    Our GNN setup closely follows the architecture and hyperparameter optimization procedure from GraphLand (Bazhenov et al., 2025), with two main differences. First, we introduce early stopping with a patience of 100 steps to accelerate training. Second, we use unified hyperparameter search spaces for both feature and target preprocessing, rather than dataset-specific spaces used in GraphLand. See our code for more details. Note that this latter modification only affects the pre-processing hyperparameters, while the search grids for learning rate and dropout remain identical to those in GraphLand.

**PEARL integration**    In the ablation studies described in Appendix C, we evaluate the effect of integrating PEARL with both GNN and LightGBM models. For GNNs, we concatenate PEARL outputs with the initial node features, and train the combined model end-to-end. For LightGBM, due to the challenge of end-to-end training with PEARL, we use the outputs from the same randomly initialized PEARL as in our G2T-FM (ICL) experiments.

## C   ABLATION

**G2T-FM components**    First, we provide an ablation of the G2T-FM components by removing them from G2T-FM and comparing performance. Here, we focus on G2T-TabPFNv2, selecting it as the representative model because TabPFNv2 is a widely adopted and well-established tabular foundation model. Table 4 shows the results of this ablation, from which we conclude that all the components are critical for the performance of G2T-FM. In particular, the following observation holds.

> **Observation 4** *Neighborhood feature aggregation (NFA) and classic structure-based features (SF) improve the overall performance of G2T-FM, while PEARL allows one to drastically improve performance in rare cases where standard augmented features are not sufficient.*

**Augmenting baselines with the same components**    Second, one may argue that the performance improvements of G2T-FM come solely from the fact that it employs augmented features that are not accessible to the GNN and LightGBM baselines. To verify this, we provide the baselines with exactly the same features as G2T-FM. The results are presented in Table 5.

Table 5: Comparison of G2T-FM against the baselines that are enhanced with the same components as G2T-FM. M stands for Modified and means that we add NFA, classic structure-based features, and PEARL encodings to their features.

| | tolokers-2 | city-reviews | artnet-exp | hm-prices | avazu-ctr | city-roads-M | twitch-views | artnet-views | **AR** |
|---|---|---|---|---|---|---|---|---|---|
| LightGBM+NFA | $56.34 \pm 0.06$ | $78.53 \pm 0.01$ | $46.13 \pm 0.03$ | $70.84 \pm 0.04$ | $31.71 \pm 0.01$ | $61.18 \pm 0.03$ | $60.14 \pm 0.01$ | $56.10 \pm 0.02$ | **7.62** |
| GCN | $56.27 \pm 0.29$ | $77.81 \pm 0.14$ | $44.86 \pm 0.34$ | $68.02 \pm 0.40$ | $32.00 \pm 0.15$ | $58.82 \pm 0.24$ | $75.51 \pm 0.05$ | $56.03 \pm 0.24$ | 9.38 |
| GraphSAGE | $54.43 \pm 0.32$ | $78.17 \pm 0.09$ | $45.14 \pm 0.34$ | $70.00 \pm 0.70$ | $31.44 \pm 0.15$ | $59.44 \pm 0.26$ | $66.29 \pm 0.31$ | $49.32 \pm 0.86$ | 10.50 |
| GAT | $57.41 \pm 0.80$ | $77.74 \pm 0.20$ | $45.06 \pm 0.49$ | $72.07 \pm 1.16$ | $32.63 \pm 0.16$ | $59.86 \pm 0.19$ | $72.89 \pm 0.25$ | $53.60 \pm 0.23$ | 7.38 |
| GT | $56.98 \pm 0.53$ | $77.34 \pm 0.20$ | $46.41 \pm 0.68$ | $69.44 \pm 0.89$ | $31.11 \pm 0.47$ | $59.55 \pm 0.27$ | $72.13 \pm 0.13$ | $53.37 \pm 0.43$ | 10.25 |
| LightGBM+NFA (M) | $57.16 \pm 0.70$ | $78.68 \pm 0.04$ | $45.57 \pm 0.19$ | $70.25 \pm 0.14$ | $31.31 \pm 0.08$ | $60.86 \pm 0.16$ | $65.17 \pm 0.04$ | $57.53 \pm 0.04$ | **7.75** |
| GCN (M) | $58.71 \pm 0.45$ | $77.07 \pm 0.27$ | $43.44 \pm 0.32$ | $70.73 \pm 0.26$ | $31.10 \pm 0.22$ | $57.91 \pm 0.22$ | $77.11 \pm 0.09$ | $56.14 \pm 0.24$ | 9.38 |
| GraphSAGE (M) | $59.59 \pm 0.51$ | $77.95 \pm 0.09$ | $44.31 \pm 0.53$ | $70.50 \pm 0.47$ | $31.51 \pm 0.41$ | $59.66 \pm 0.09$ | $75.93 \pm 0.19$ | $55.39 \pm 0.32$ | 7.75 |
| GAT (M) | $57.76 \pm 0.70$ | $77.47 \pm 0.14$ | $44.36 \pm 0.50$ | $72.46 \pm 0.49$ | $31.97 \pm 0.23$ | $59.57 \pm 0.43$ | $77.20 \pm 0.18$ | $56.51 \pm 0.35$ | 6.62 |
| GT (M) | $58.79 \pm 0.76$ | $76.43 \pm 0.10$ | $43.03 \pm 0.60$ | $71.84 \pm 0.64$ | $29.86 \pm 0.67$ | $59.85 \pm 0.41$ | $76.15 \pm 0.11$ | $56.39 \pm 0.31$ | 8.50 |
| G2T-TabPFNv2 (ICL) | $60.42 \pm 0.27$ | $77.46 \pm 0.10$ | $45.84 \pm 0.03$ | $66.68 \pm 0.09$ | $26.38 \pm 0.07$ | $60.47 \pm 0.04$ | $70.00 \pm 0.06$ | $58.75 \pm 0.15$ | 8.62 |
| G2T-TabPFNv2 (FT) | $57.65 \pm 1.92$ | $79.12 \pm 0.21$ | $47.31 \pm 0.59$ | $71.05 \pm 0.91$ | $28.52 \pm 0.43$ | $63.08 \pm 0.28$ | $74.06 \pm 0.16$ | $60.29 \pm 0.13$ | 5.50 |
| G2T-LimiX (ICL) | $61.48 \pm 0.30$ | $77.72 \pm 0.54$ | $48.43 \pm 0.18$ | $74.96 \pm 0.06$ | $32.39 \pm 0.14$ | $64.53 \pm 0.07$ | $71.08 \pm 0.07$ | $60.95 \pm 0.10$ | 3.88 |
| G2T-LimiX (FT) | $61.17 \pm 0.49$ | $80.13 \pm 0.05$ | $49.88 \pm 0.13$ | $76.32 \pm 0.17$ | $33.94 \pm 0.34$ | $65.87 \pm 0.10$ | $73.16 \pm 0.40$ | $62.12 \pm 0.10$ | 1.88 |

> **Observation 5** *While some improvements achieved by G2T-FM can be explained by its access to the features that are not used by traditional GNNs, G2T-FM shows strong performance even against the enhanced baselines. In particular, on some datasets it outperforms all other methods by a wide margin.*

**Summary** To sum up, the gains of G2T-FM come from the synergy between the TFM backbone and our graph-to-table components. The ablations show that removing any component degrades the performance, and providing the baselines with the same augmented features does not close the gap. Hence, both the backbone and the proposed components are necessary for the strong performance.

# D  SYMMETRIES: EQUIVARIANCE AND INVARIANCE

For graph problems, it is typically assumed that node IDs can be relabeled, feature columns can be reordered, and class IDs can be renamed without changing the underlying task. Hence, a model should not depend on these arbitrary choices. This motivates three symmetries for graph foundation models, as advocated in Finkelshtein et al. (2025): (i) feature permutation invariance; (ii) label permutation equivariance; and (iii) node permutation equivariance.

Formally, let $G$ be a group acting on inputs $\mathcal{X}$ and outputs $\mathcal{Y}$. A mapping $f : \mathcal{X} \to \mathcal{Y}$ is $G$-equivariant if $f(g \cdot x) = g \cdot f(x)$ for all $g \in G$, $x \in \mathcal{X}$. It is $G$-invariant if $f(g \cdot x) = f(x)$ for all $g$. In our context, relevant groups include node permutations $S_{|V|}$, feature (column) permutations $S_d$, and label permutations $S_{|C|}$. Here, $S_n$ denotes the symmetric group on a set of $n$ elements (i.e., the group of all permutations), $|V|$ is the number of nodes, $d$ is the number of features, and $|C|$ is the number of classes.

Modern models may also include stochastic components (e.g., random positional encodings (Kanatsoulis et al., 2025)). In such cases, one may require symmetries to hold *in distribution*: after the relevant permutation, the distribution of outputs is unchanged (for invariance) or transformed accordingly (for equivariance), even if a single stochastic realization is not exactly symmetric.

Below, we discuss the symmetries of G2T-FM. In particular, we show that all components added to TFM are equivariant in distribution, so the resulting symmetries of G2T-FM depend on the symmetries of the TFM backbone. As an example, we analyze the symmetries of TabPFNv2.

**G2T-FM components** The symmetries of G2T-FM rely on the symmetries of the chosen TFM. Let us show that if the TFM has feature permutation invariance, label permutation equivariance and sample permutation equivariance (in distribution), then G2T-FM also has all the desired symmetries. Indeed, G2T-FM employs several components: NFA, PEARL, and simple structure-based features (degree, PageRank, Laplacian eigenvectors). NFA is node- and feature-equivariant. Structural features such as node degree, PageRank, and Laplacian eigenvectors are all node-equivariant by construction. The PEARL framework, which we also use, is node-equivariant *in distribution*.

The combination of these components makes G2T-FM node permutation equivariant *in distribution*. Thus, all the desired symmetries are preserved.

**TabPFNv2 backbone** TabPFNv2 is sample permutation equivariant and feature permutation invariant *in distribution*. This property holds because the model applies random positional encodings to its input features. As these encodings are sampled independently and identically, the output distribution is unaffected by the order of the feature columns. By default, TabPFNv2 is not label permutation equivariant. However, this can be achieved with a simple modification that we incorporate into our implementation. Specifically, during each forward pass, we randomly permute the ordinal encodings assigned to the class labels. This procedure ensures that the model becomes label permutation equivariant *in distribution*.

## E   LLM USAGE

LLMs have been used for proofreading and minor stylistic editing of the paper; the authors are responsible for all the content.

