# OpenReview forum: "Turning Tabular Foundation Models into Graph Foundation Models"
_ICLR.cc/2026/Conference — Submitted to ICLR 2026_

### Official Review · Reviewer_EACW · 2025-10-23

**Soundness:** 2
**Presentation:** 2
**Contribution:** 2
**Rating:** 2
**Confidence:** 4

**Summary:**

This paper introduces G2T-FM, which turns graphs into enriched tabular data by concatenating (i) neighborhood feature aggregations, (ii) classic structural stats (degree, PageRank, Laplacian eigenvectors), and (iii) PEARL-style structural encodings, then feeds this into a tabular foundation model (TabPFNv2 or LimiX). In in-context and finetuning regimes it reports competitive or superior node-level performance to tuned GNNs and clearly stronger results than publicly available GFMs; the approach also supports regression.

**Strengths:**

1. It is a novel approach to apply tabular foundation model to the graph domain.
2. It unifies both the regression and classification of node property prediction task in one framework.

**Weaknesses:**

1. The method’s graph learning capacity largely relies on the tabular foundation model (TFM) backbone. The paper does not analyze the backbone’s inductive biases or provide any “ability boundary” characterization (e.g., what classes of structural patterns can/can’t be captured), leaving the effective limits of the approach unclear.
2. Computing Laplacian eigenvectors and PEARL embeddings can be expensive on large graphs, yet there is no end-to-end complexity or runtime/memory analysis.
3. The framework uses only basic structural processing; there is no cross-graph pretraining or learned multi-hop structural module. As a result, tasks relying on the structures may remain underperforming.
4. Despite the novel use of TFMs for graphs, experiments are confined to transductive node-level prediction. Without evidence of inductive generalization (new nodes/graphs), cross-graph transfer, and multi-task coverage (edge/graph-level), the “graph foundation model” claim feels premature.

**Questions:**

1. How was the finetuning of the G2T-TabPFNv2/G2T-LimiX backbone of conducted? Can more details be provided?
2. How does G2T-FM perform when evaluated inductively (new nodes/graphs at test time without transductive access)? Any changes needed?
3. What are the time/memory costs for NFA, Laplacian eigenvectors, and PEARL as node/edge counts grow? Could you provide wall-clock and peak-memory vs. GNN baselines?
4.How sensitive are results to the number of Laplacian components, PEARL repeats, and NFA hop/aggregation choices? Can you report per-dataset optima/robustness?
5. Can the framework be applied to other graph learning tasks? What kinds of changes are needed?

---

> ### Author Response · Authors · 2025-11-19
> **Official Comment (Part 1/5)**
>
> Thank you for your review. We address your questions and comments below. Please also see the general response where we share our view on the positioning of this work in the GFM field.
>
> > **W1**. The method’s graph learning capacity largely relies on the tabular foundation model (TFM) backbone. The paper does not analyze the backbone’s inductive biases or provide any “ability boundary” characterization (e.g., what classes of structural patterns can/can’t be captured), leaving the effective limits of the approach unclear.
>
> Let us note that the considered backbones are examples of the Prior-data Fitted Networks (PFNs), which is a relatively young approach and the subject of active research now. We believe that deeper analysis of its inductive biases will appear in the near future, but at this point the understanding of its benefits and limits is largely derived from its empirical performance. Similarly, in our work, we show that this framework can be effectively applied to realistic tasks from the graph domain.
>
> Regarding the ability boundary of our graph processing, as we discuss in lines 707-712, we use only basic methods. In particular, the NFA approach only aggregates features from 1-hop neighbors and thus may not capture complex longer-range graph-feature dependencies. However, even such a simple approach proves to be sufficient to show strong performance. On the other hand, the usage of PEARL allows us to effectively process the graph structure: [3] claims that PEARL surpasses the expressiveness of the Weisfeiler-Leman test and is capable of counting some important substructures at the node level. These properties are inherited by our framework.
>
> > **W2**. Computing Laplacian eigenvectors and PEARL embeddings can be expensive on large graphs, yet there is no end-to-end complexity or runtime/memory analysis.
>
> > **Q3**. What are the time/memory costs for NFA, Laplacian eigenvectors, and PEARL as node/edge counts grow? Could you provide wall-clock and peak-memory vs. GNN baselines?
>
> In this work, we aimed to answer the general question on whether it is possible to design a GFM that can compete in terms of predictive performance with well-tuned GNNs on arbitrary tasks in realistic scenarios. Our approach allows us to give a positive answer to this question, and we leave scaling to larger graphs for future research (please see our general response for a more detailed discussion).
>
> To address your comment and for the completeness of our study, we are currently benchmarking the efficiency of our approach and will provide the results when the experiments are finished.
>
> For now, below we analyze the time/space complexity of our method. We use the following notation: $N$ stands for the number of nodes, $E$ for the number of edges and $F$ for the number of features. For convenience, we also assume that $E + 1 \geq N$, which always holds for connected graphs. For conciseness, we assume all other hyperparameters to be fixed and do not include them in $O$-notation.
>
> The graph preprocessing step consists of computing NFA, LapPE, degree, PageRank and PEARL embeddings.
> - NFA has $O(EF)$ time complexity and $O(NF)$ space complexity.
> - Computing a fixed number of LapPE features requires $O(E)$ time and space complexity.
> - Computing degree has $O(E)$ time and $O(N)$ memory complexity.
> - Computing PageRank requires $O(E)$ time and $O(N)$ space complexity for each iteration, and we use at most 100 iterations.
> - PEARL is essentially a GNN applied several times, so it has $O(E)$ time complexity and $O(N)$ space complexity.
>
> Overall, the preprocessing has $O(EF)$ time complexity and $O(NF+E)$ space complexity.
>
> Since both TabPFNv2 and LimiX have dual attention (i.e., interleaving attention operations over sample dimension and over feature dimension), they have $O(N^2 F + N F^2)$ time complexity and $O(NF)$ memory complexity.

---

> ### Author Response · Authors · 2025-11-19
> **Official Comment (Part 2/5)**
>
> > **W3**. The framework uses only basic structural processing; there is no cross-graph pretraining or learned multi-hop structural module. As a result, tasks relying on the structures may remain underperforming.
>
> We believe that the lack of cross-dataset pretraining is actually an advantage of our method. Specifically, it can be immediately applied to any existing or future TFM without costly graph-specific pretraining. Therefore, while cross-graph pretraining is an important next step, we leave it for future research.
>
> Also, regarding the absence of multi-hop structural modules, let us note that the PEARL module used in our framework is essentially a learnable multi-hop structural module. Thus, tasks relying solely on graph structure can potentially be learned. On the other hand, for NFA, we indeed use 1-hop aggregations. We do not go beyond that since naive stacking of NFAs over multiple hops would significantly increase the number of features, and both TabPFNv2 and LimiX can handle only a limited number of features. However, our evaluation results show that even this simple approach already allows G2T-FM to outperform well-tuned GNNs trained for specific datasets. Thus, while using learnable multi-hop aggregations can lead to better performance, we consider this a direction for future work.
>
> > **W4**. Despite the novel use of TFMs for graphs, experiments are confined to transductive node-level prediction. Without evidence of inductive generalization (new nodes/graphs), cross-graph transfer, and multi-task coverage (edge/graph-level), the “graph foundation model” claim feels premature.
>
> We discuss the performance of our approach on inductive tasks below (see our response to Q2). Regarding the limitations of our approach and the positioning of our work within the GFM literature, we have added a paragraph in the Limitations section (Appendix A) to clearly outline the scope of our work and avoid possible confusion. In particular, we note that in the graph ML community there is no clear definition of a GFM and there are several previously proposed task-specific (e.g., node-level only) GFMs, such as TS-GNN, GraphFM and GCOPE.
>
> > **Q1**. How was the finetuning of the G2T-TabPFNv2/G2T-LimiX backbone of conducted? Can more details be provided?
>
> As we state in line 738, we closely follow the setup of Rubachev et al. (2025) for finetuning. Specifically, we conduct full finetuning of both the TFM backbone and PEARL. We use the Adam optimizer and tune the learning rate on the validation set using a logspace grid with 10 learning rate values ranging from `5e-6` to `5e-4`. We do not set a maximal number of finetuning steps, but instead use early stopping. Namely, we compute the performance on the validation set every ten steps and stop finetuning after 16 non-improving evaluations. We are happy to provide additional details if anything remains unclear.

---

> ### Author Response · Authors · 2025-11-19
> **Official Comment (Part 3/5)**
>
> > **Q2**. How does G2T-FM perform when evaluated inductively (new nodes/graphs at test time without transductive access)? Any changes needed?
>
> First, we would like to note that for G2T-FM (ICL) there is no difference between transductive and inductive setups, since the model makes predictions in a single forward pass without any training. We believe this can be a significant advantage of ICL models.
>
> Second, we evaluate G2T-LimiX (ICL) on GraphLand THI (temporal high inductive) splits and compare it with other baselines. In the THI setup, nodes are splitted into train/validation/test according to the time attribute, and the validation/test nodes are not seen during training. It is important to note that, in addition to the inductive problem setup, THI split creates a noticeable time-based distribution shift. However, **most existing TFMs are pretrained under the IID assumption**. Hence, we believe one should not expect that they will achieve SOTA results.
>
> The results show that G2T-LimiX (ICL) consistently outperforms prior GFMs. Surprisingly, on some datasets (`artnet-views`, `hm-prices`) G2T-LimiX (ICL) achieves SOTA or nearly-SOTA results even despite distribution shifts. On the remaining datasets, its performance is competitive or even worse than that for task-specific baselines trained from scratch.
>
> However, we would like to note that, while most TFMs are pretrained under the IID assumption, one can incorporate distribution shift awareness into their pretraining process, and [1] shows that this can boost TFM performance in OOD scenarios. Incorporating such approaches into GFMs could be a promising direction for future work.
>
> |                 | tolokers-2    | artnet-exp   | hm-prices    | avazu-ctr    | twitch-views | artnet-views |
> | :-------------- | :------------ | :----------- | :----------- | :----------- | :----------- | :----------- |
> | LightGBM-NFA    | 45.45 ± 0.82  | 39.91 ± 0.25 | 68.88 ± 0.11 | 33.04 ± 0.17 | 24.81 ± 0.11 | 51.95 ± 0.05 |
> | GCN             | 32.43 ± 8.03  | 41.28 ± 0.28 | 64.31 ± 0.82 | 34.78 ± 0.48 | 63.58 ± 0.54 | 53.73 ± 0.47 |
> | GraphSAGE       | 30.86 ± 9.48  | 40.53 ± 0.40 | 65.80 ± 0.56 | 36.79 ± 0.55 | 56.60 ± 0.71 | 53.37 ± 0.30 |
> | GAT             | 24.53 ± 9.55  | 41.64 ± 0.32 | 69.74 ± 1.50 | 37.18 ± 1.02 | 61.41 ± 1.30 | 52.44 ± 0.59 |
> | GT              | 22.89 ± 10.40 | 40.26 ± 0.82 | 67.33 ± 2.05 | 36.49 ± 0.83 | 60.67 ± 1.02 | 52.26 ± 0.48 |
> | OpenGraph (ICL) | 9.12 ± 1.74   | 16.19 ± 1.36 | N/A          | N/A          | N/A          | N/A          |
> | AnyGraph (ICL)  | 13.52 ± 4.74  | 11.80 ± 1.01 | N/A          | N/A          | N/A          | N/A          |
> | TS-GNN (ICL)    | 12.59 ± 1.97  | 9.46 ± 1.25  | N/A          | N/A          | N/A          | N/A          |
> | GCOPE (FT)      | 10.73 ± 1.48  | 19.10 ± 0.96 | N/A          | N/A          | N/A          | N/A          |
> | G2T-LimiX (ICL) | 34.98 ± 3.87  | 22.79 ± 0.45 | 69.58 ± 0.12 | 36.15 ± 1.54 | 58.26 ± 0.38 | 54.56 ± 0.20 |

---

> ### Author Response · Authors · 2025-11-19
> **Official Comment (Part 4/5)**
>
> > **Q4.1**. How sensitive are results to the number of Laplacian components, PEARL repeats, and NFA hop/aggregation choices? Can you report per-dataset optima/robustness?
>
> Thank you for this suggestion! We have conducted additional experiments and provide their results below. Also, let us clarify that in our main experiments we fixed all the hyperparameters mentioned above and did not tune them for each dataset individually.
>
> **LapPE**: We evaluated G2T-LimiX with a varying number of LapPE components, while fixing all other hyperparameters to the default ones. The results are presented in the table below. One can see that for certain datasets (e.g., `tolokers-2` or `artnet-views`) adding more components consistently improves the performance, offering a tradeoff between downstream performance and time/memory-efficiency. However, on some other datasets (e.g., `city-reviews` or `avazu-ctr`) adding more components either does not affect performance or even makes it worse. Overall, we believe that 14-16 components is an adequate default value.
>
> |          | tolokers-2   | city-reviews | artnet-exp   | hm-prices    | avazu-ctr    | city-roads-M | twitch-views | artnet-views |
> | :------- | :----------- | :----------- | :----------- | :----------- | :----------- | :----------- | :----------- | :----------- |
> | LapPE 00 | 57.75 ± 0.11 | 78.73 ± 0.54 | 48.13 ± 0.10 | 74.82 ± 0.02 | 32.70 ± 0.06 | 65.37 ± 0.10 | 68.92 ± 0.01 | 58.60 ± 0.02 |
> | LapPE 02 | 60.61 ± 0.23 | 78.73 ± 0.28 | 48.14 ± 0.24 | 74.84 ± 0.05 | 32.71 ± 0.11 | 65.17 ± 0.07 | 69.89 ± 0.16 | 58.82 ± 0.04 |
> | LapPE 04 | 61.01 ± 0.22 | 78.49 ± 0.33 | 48.36 ± 0.08 | 74.84 ± 0.05 | 32.60 ± 0.17 | 65.05 ± 0.11 | 70.33 ± 0.04 | 59.26 ± 0.06 |
> | LapPE 08 | 61.64 ± 0.21 | 78.39 ± 0.37 | 48.36 ± 0.21 | 74.94 ± 0.04 | 32.65 ± 0.15 | 64.70 ± 0.11 | 70.70 ± 0.08 | 60.19 ± 0.06 |
> | LapPE 16 | 61.54 ± 0.17 | 77.71 ± 0.60 | 48.54 ± 0.21 | 74.91 ± 0.04 | 32.16 ± 0.20 | 64.65 ± 0.10 | 71.25 ± 0.07 | 61.03 ± 0.08 |
> | LapPE 32 | 61.78 ± 0.24 | 78.05 ± 0.67 | 48.79 ± 0.24 | 75.01 ± 0.16 | 30.54 ± 0.44 | 64.60 ± 0.05 | 71.46 ± 0.06 | 61.61 ± 0.09 |
> | LapPE 64 | 62.09 ± 0.16 | 78.19 ± 0.63 | 49.74 ± 0.17 | 75.89 ± 0.11 | 28.80 ± 1.88 | 64.58 ± 0.09 | 71.61 ± 0.09 | 61.99 ± 0.05 |
>
> **PEARL**: We evaluated G2T-LimiX with a varying number of PEARL repeats, while fixing all other hyperparameters to their default values. The results are presented in the table below. One can see that the number of PEARL repeats has a minimal effect on the downstream performance.
>
> |          | tolokers-2   | city-reviews | artnet-exp   | hm-prices    | avazu-ctr    | city-roads-M | twitch-views | artnet-views |
> | :------- | :----------- | :----------- | :----------- | :----------- | :----------- | :----------- | :----------- | :----------- |
> | PEARL 01 | 61.46 ± 0.28 | 77.87 ± 0.51 | 48.49 ± 0.16 | 74.96 ± 0.06 | 32.34 ± 0.18 | 64.54 ± 0.07 | 70.72 ± 0.05 | 60.95 ± 0.08 |
> | PEARL 02 | 61.51 ± 0.28 | 77.78 ± 0.57 | 48.47 ± 0.18 | 74.97 ± 0.06 | 32.37 ± 0.12 | 64.54 ± 0.10 | 70.83 ± 0.06 | 60.96 ± 0.09 |
> | PEARL 04 | 61.52 ± 0.26 | 77.77 ± 0.46 | 48.51 ± 0.12 | 74.97 ± 0.06 | 32.41 ± 0.14 | 64.50 ± 0.11 | 70.99 ± 0.07 | 60.95 ± 0.12 |
> | PEARL 08 | 61.48 ± 0.30 | 77.72 ± 0.54 | 48.43 ± 0.18 | 74.96 ± 0.06 | 32.39 ± 0.14 | 64.53 ± 0.07 | 71.08 ± 0.07 | 60.95 ± 0.10 |
> | PEARL 16 | 61.49 ± 0.32 | 77.75 ± 0.62 | 48.44 ± 0.18 | 74.96 ± 0.06 | 32.35 ± 0.21 | 64.54 ± 0.08 | 71.13 ± 0.07 | 60.94 ± 0.10 |
> | PEARL 32 | 61.44 ± 0.34 | 77.73 ± 0.56 | 48.49 ± 0.14 | 74.95 ± 0.07 | 32.34 ± 0.23 | 64.48 ± 0.08 | 71.16 ± 0.05 | 60.95 ± 0.10 |
>
> **NFA**: Please note that in all our experiments we used 1-hop NFA since naively adding more hops would significantly increase the number of features for TFM. Therefore, we only analyzed the effect of different aggregations. We evaluated a variant of G2T-LimiX which uses only mean-aggregation for all features and compared it with the default one, which additionally aggregates numerical features with max- and min-aggregations. The results are presented in the table below. Overall, using only mean-aggregation seems to have a minimal effect on the downstream performance, yet there is a drop on the `city-reviews` dataset, so we believe using all aggregations is a more robust solution.
>
> |                | tolokers-2   | city-reviews | artnet-exp   | hm-prices    | avazu-ctr    | city-roads-M | twitch-views | artnet-views |
> | :------------- | :----------- | :----------- | :----------- | :----------- | :----------- | :----------- | :----------- | :----------- |
> | mean, max, min | 61.48 ± 0.30 | 77.72 ± 0.54 | 48.43 ± 0.18 | 74.96 ± 0.06 | 32.39 ± 0.14 | 64.53 ± 0.07 | 71.08 ± 0.07 | 60.95 ± 0.10 |
> | mean           | 61.72 ± 0.21 | 76.84 ± 0.37 | 48.76 ± 0.07 | 74.83 ± 0.07 | 32.46 ± 0.14 | 64.69 ± 0.12 | 71.05 ± 0.05 | 60.95 ± 0.06 |

---

> > ### Author Response · Authors · 2025-11-19
> > **Official Comment (Part 5/5)**
> >
> > > **Q4.2**. Can the framework be applied to other graph learning tasks? What kinds of changes are needed?
> >
> > Potentially, G2T-FM can be applied to link-level tasks in the following way. First, one applies the same preprocessing steps (like NFA) of G2T-FM, similar to node-level tasks. Second, for any pair of nodes one can set the target to one, if an edge is present in the graph, and zero otherwise. Features for a pair of nodes are obtained by the concatenation of the nodes’ features. Third, one can employ a negative sampling procedure similar to classic link prediction to form a set of negative examples. Finally, one can apply a TFM to the obtained dataset to make ICL predictions for new node pairs. For PEARL, one can also take inspiration from [2] on how to better use it for link-level tasks.
> >
> > For graph-level tasks, we do not see an elegant way to apply G2T-FM. We believe that graph-level tasks are very different from node- and link-level tasks and thus may require distinct foundation models. Nevertheless, as we discuss in our general response, creating a GFM that can compete with GNNs in a realistic node-level setup is a challenging task and we believe that our contribution is valuable.
> >
> > ***
> >
> > [1] Drift-Resilient TabPFN: In-Context Learning Temporal Distribution Shifts on Tabular Data, NeurIPS 2024
> >
> > [2] Holographic Node Representations: Pre-training Task-Agnostic Node Embeddings, ICLR 2025.
> >
> > [3] Learning Efficient Positional Encodings with Graph Neural Networks, ICLR 2025

---

> > > ### Author Response · Authors · 2025-11-26
> > >
> > > Dear Reviewer EACW,
> > >
> > > We have recently completed our experiments on benchmarking time and memory efficiency of considered methods, please refer to [this comment](https://openreview.net/forum?id=8tW32RrecK&noteId=SZlXIhNt0D) for the results. You might also find interesting our recent results on incorporating multi-hop NFA to G2T-FM framework, see [this comment](https://openreview.net/forum?id=8tW32RrecK&noteId=zAP4Wwuaqr). In short, potential efficiency problems of G2T-FM seem to be not so evident as one may think, and adding multi-hop NFA does not lead to consistent improvements. Please let us know if you have any remaining concerns, we would be happy to discuss them further. Following our [general response](https://openreview.net/forum?id=8tW32RrecK&noteId=itqnPkUCKC), we would also be happy to discuss positioning of this work in the GFM field and future potential of graph foundation models in general.
> > >
> > > Best regards,
> > >
> > > Authors

---

### Official Review · Reviewer_Cmbg · 2025-10-28

**Soundness:** 3
**Presentation:** 4
**Contribution:** 3
**Rating:** 6
**Confidence:** 4

**Summary:**

This paper proposes G2T-FM, a simple framework that adapts tabular foundation models (TFMs) to graph machine learning, addressing the challenge of heterogeneous node feature spaces and target spaces. Experiments on diverse datasets show that G2T-FM in both in-context and finetuning regimes can outperform well-tuned GNN baselines and existing openly available GFMs.

**Strengths:**

1. The paper introduces a clear and compelling analogy between tabular data and heterogeneous graph features, enabling the transfer of advances from tabular foundation models (TFMs) into the graph machine learning domain.

2. It provides a thorough and insightful discussion of the limitations of prior graph foundation models, highlighting gaps in generalization and feature handling.

3. On multiple datasets, the proposed approach achieves competitive or superior results compared to well-tuned traditional GNN baselines and existing GFM implementations, despite its simplicity.

**Weaknesses:**

1. The idea is insightful but the technical realization is relatively minimal. The method consists largely of straightforward one-hop feature aggregation and concatenation followed by application of an existing TFM.

2. The framework aggregates only one-hop neighbor node features, raising concerns about its expressiveness and ability to capture more complex, multi-hop dependencies.

3. Certain experimental settings and results require further justification. Specifically: Can TS-GNN actually be finetuned (line 370)? Why does G2T-LimiX (ICL) outperform G2T-LimiX (FT) on the tolokers-2 dataset (Table 2)? Why were different data splits used (line 415) compared to other GFMs?

**Questions:**

1. Why not go further and pretrain a dedicated GFM based on your proposed graph-to-tabular framework? For example, converting the large-scale graph datasets typically used to train GFMs into tabular form and performing cross-graph pretraining could yield a more impactful contribution.

2. See Weaknesses 3.

---

> ### Author Response · Authors · 2025-11-19
> **Official Comment (Part 1/2)**
>
> Thank you for your review and support of our work. We address your questions and comments below. Please also see the general response where we share our view on the positioning of this work in the GFM field.
>
> > **W1**. The idea is insightful but the technical realization is relatively minimal. The method consists largely of straightforward 1-hop feature aggregation and concatenation followed by application of an existing TFM.
>
> We consider the technical simplicity of our approach to be its important advantage. It is easy to implement and can be used as an important reference point in future works developing more powerful foundation models. Given that the field of TFMs is rapidly developing, we consider it to be an advantage that G2T-FM can be easily combined with any current or future TFM.
>
> > **W2**. The framework aggregates only 1-hop neighbor node features, raising concerns about its expressiveness and ability to capture more complex, multi-hop dependencies.
>
> We acknowledge that in our current implementation, the neighborhood feature aggregation is fixed and limited to 1-hop neighbors. We do not go beyond that since naive stacking of NFAs over multiple hops would significantly increase the number of features, and both TabPFNv2 and LimiX can handle a limited number of features. However, our evaluation results show that even this simple approach already allows G2T-FM to outperform well-tuned GNNs trained for specific datasets. Also note that we employ the PEARL component, which allows us to potentially capture more complex structural patterns and cover multi-hop dependencies. So, while using learnable multi-hop aggregations can lead to better performance, we leave it to future research.
>
> > **W3** and **Q2**. Certain experimental settings and results require further justification. Specifically:
>
> > Can TS-GNN actually be finetuned (line 370)?
>
> It is true that any neural model, including AnyGraph, OpenGraph and TS-GNN, can actually be finetuned on a downstream task. However, these particular models are presented by their authors as zero-shot prediction (or, using more appropriate terminology, in-context learning) methods, and their current official implementations do not support finetuning. Thus, we used exactly the setting in which these models are intended to be tested.
>
> > Why does G2T-LimiX (ICL) outperform G2T-LimiX (FT) on the tolokers-2 dataset (Table 2)?
>
> Indeed, we found that neither G2T-TabPFNv2 nor G2T-LimiX benefits from finetuning on `tolokers-2`. This may be attributed to randomness in the learning process or overfitting. However, we believe that this fact does not affect the general observations presented in our paper.
>
> > Why were different data splits used (line 415) compared to other GFMs?
>
> We chose not to use few-shot data splits for several reasons. First, we believe that few-shot settings are less realistic for most practical applications. Second, few-shot splits for GNNs often produce high variance and can disadvantage GNN baselines as they are trained from scratch. Thus, we used larger splits to examine whether GFMs can compete with strong, well-tuned GNNs in a regime where GNNs are known to perform well.

---

> > ### Author Response · Authors · 2025-11-19
> > **Official Comment (Part 2/2)**
> >
> > > **Q1**. Why not go further and pretrain a dedicated GFM based on your proposed graph-to-tabular framework? For example, converting the large-scale graph datasets typically used to train GFMs into tabular form and performing cross-graph pretraining could yield a more impactful contribution.
> >
> > Thank you for this valuable suggestion. We agree that cross-graph pretraining is an important direction for future research and acknowledge its potential impact. However, we believe that G2T-FM occupies a complementary space, and that its ability to use any TFM without any graph-specific pretraining represents an important contribution in its own right.
> >
> > The core design principle of G2T-FM is to allow simple and direct application to any present or future TFM. This is enabled by our approach, which does not require expensive or complex pretraining specifically for graphs. Given the rapid development of the TFM field, this flexibility and ease of integration are significant advantages, as they ensure that G2T-FM can automatically benefit from further improvements in tabular models.
> >
> > Pretraining a dedicated GFM via cross-graph objectives would indeed be valuable, but such a direction would likely require substantial modifications to our current framework. For instance, one could replace the learning-free, hand-crafted features in G2T-FM with learnable message-passing layers, resulting in a fundamentally different model.
> >
> > In addition, we believe that, following the PFN framework, it might be more appropriate to pretrain on synthetic graphs rather than on real-world ones. However, designing a generator for such synthetic graph data is a non-trivial task and would introduce significant additional complexity.
> >
> > For these reasons, we see cross-graph pretraining as a promising yet separate line of future work. In this paper, our focus is on demonstrating the effectiveness of the simple, plug-and-play G2T-FM framework, which can be easily combined with any TFM without the need for graph-specific pretraining. We leave cross-graph pretraining to future research.

---

> > > ### Comment · Reviewer_Cmbg · 2025-11-25
> > >
> > > I thank the authors for their response. While I appreciate the novel methodology and will maintain my positive assessment, my primary concerns regarding W1 and W2 remain unaddressed. Consequently, I cannot raise my rating further.

---

> > > > ### Author Response · Authors · 2025-11-26
> > > >
> > > > Dear Reviewer Cmbg,
> > > >
> > > > Thank you for your reply and positive assessment! Following your comments W1 and W2, we have conducted additional experiments with multi-hop aggregations. In short, they do not lead to consistent performance improvements, so we believe one-hop aggregations serve as a strong baseline. Please see [this comment](https://openreview.net/forum?id=8tW32RrecK&noteId=zAP4Wwuaqr) for details. If you have any additional comments or suggestions, we would be happy to discuss further.
> > > >
> > > > Best regards,
> > > >
> > > > Authors

---

### Official Review · Reviewer_uY6A · 2025-10-29

**Soundness:** 3
**Presentation:** 2
**Contribution:** 2
**Rating:** 4
**Confidence:** 4

**Summary:**

The authors propose a preprocessing pipeline for graph learning tasks to convert them to tabular form effectively such that tabular foundation models (TFM) can operate on them. Their pipeline consists of a mixture of hand-crafted, heuristic and learnable feature extractors to obtain tabular features, which is fed into a TFM in either an in-context-learning or fine-tuning setting. The authors demonstrate competitive performance across several benchmarks in which they comfortably outperform graph foundation models (GFM) and are competitive against a variety of GNNs.

**Strengths:**

1. The paper is well-written; the contributions are clearly stated and the logical flow is easy to follow.
2. The core idea of leveraging well-established tools from graph learning literature to adapt graph tasks to TFMs, which is conceptually simple but effective.
3. The results clearly back the authors’ claims; ICL results on datasets with text-based features are particularly promising. I am convinced that G2T-FMs work reasonably well even in an ICL setting, but require further convincing on the overall usefulness over existing methods (see Weaknesses).

**Weaknesses:**

1. I agree that the ability to leverage TFMs for graph tasks is a valuable contribution in itself, but I don’t think the paper’s contributions go much beyond that, resulting in a paper limited in scope and largely relying on the success of its implementation. In relation with this, I think the paper somewhat oversells its contributions -- while I understand the reasoning to associate the resulting framework with graph foundation models (GFM), I think it’s bit of a stretch to argue that the resulting model is a GFM in the conventional sense. Applying the proposed framework to _any_ graph requires pre-computing structure-based features, which comes with a non-trivial computational cost per graph; avoiding such hand-crafted feature engineering is one of the main driving forces of graph representation learning, and in a related manner GFMs in the first place. The hand-crafted features alone mean the learned embeddings are not transferable by themselves without computing these features for unseen graphs first.
2. Weak benchmarks: The current experimental section needs to be significantly strengthened to make a more convincing argument towards the merits of G2T-FMs. The crux of the paper is that using tools like NFA, structural features and heuristic (LapPE)/learnable (PEARL) positional encodings allow us to apply TFMs in graph data. Note that most if not all of these tools can be directly applied to not just G2T-FMs but also both GNN and GFN benchmarks compared against. Measuring G2T-FMs against benchmark methods that do not also use structural features or positional encodings results in unfair evaluation — whether G2T-FMs can outperform other architectures when the same structural information is provided to all will provide a much healthier signal on the usefulness of TFMs on graph data. I suggest several evaluation settings to the authors in the Questions section.
3. In relation to the previous point, the authors don’t address how their G2T-FMs compare with the other benchmarks, in particular the GFMs, in terms of efficiency.  What are the parameter counts for the optimal models? How long does pre-training and/or fine-tuning take for each? How fast is downstream inference? Answering these questions will allow evaluating the merits of the proposed model better, but the information simply isn’t there. Similarly, the cost of graph pre-processing is not discussed, which is crucial considering they compare against benchmarks that do not have this pre-processing overhead.

**Additional comments (no effect on score):**
- I think referring to the Shi et al. (2021) architecture as GT in shorthand in the experiments is confusing since it uses local attention as opposed to global attention over the graph (more akin to GAT in this), which is typically considered the defining characteristic of GT architectures; Shi et al. (2021) themselves refer to their model as UniMP so I suggest reverting to that.

**Conclusion:** I think this work is primarily a method paper with relatively small theoretical component — and this is to an extent fine, with the caveat that the potential impact of the paper will then be largely determined by whether it provides any performance or efficiency gains in competitive scenarios. Thus, my view is that for acceptance this work needs to be _very_ convincing regarding these performance or efficiency gains; with the current evaluations, while promising, I am not fully convinced this is the case (hence my focus on the weaknesses in evaluation and request for additional results, something I try to avoid asking unless well-justified). Therefore I currently recommend rejection, though again with better evaluation and convincing results I may be persuaded.

**Questions:**

1. Re: W2, Here are several setups that I would have liked to see G2T-FMs compete against:
   - GNNs with (a) structural features, (b) heuristic positional/structural encodings (PSE) like Laplacian PE and andom walk encodings (RWSE), and (c) learnable PSEs like GPSE [1] and PEARL. At the very least, GNN results using identical features & encodings (namely, node degree, PageRank, Laplacian PE, PEARL, _and_ their combinations) are required. I suggest RWSE on the basis that it may capture different structural information than Laplacian PEs (in the sense that they may complement each other); I suggest GPSE because it is a _learnable_ PSEs similar to PEARL, but learns over a large variety of PSEs to arrive at a unified representation and demonstrates generalization capabilities over OOD graphs.
   - _Global_ graph transformer (GT) architectures with the above structural features & PSEs. With global, I refer to GTs that leverage _non-local_ attention, unlike GAT or the Shi et al. (2021) GT. Of course, quadratic scaling of GTs on large graphs pose a problem here, so sparse GT implementations like Performer [2]/Exphormer [3]/NodeFormer [4] etc. would be more appropriate here. I suggest picking one architecture and focusing on a subset of more heterophilic tasks where GTs are more likely to outperform GNNs.
   - GFN benchmarks with the above structural features & PSEs. These GFNs should be able to handle arbitrary node features, _and_ at least some of them can likely benefit from such structural information akin to conventional GNN/GTs.
2. Re: W3, as mentioned in the Weaknesses section, I suggest the authors provide information on (a) model size and pre-training/fine-tuning/evaluation efficiency, (b) pre-processing overhead of the G2T-FM pipeline, and provide an overview of the benefits of G2T-FMs from this standpoint.

[1] Cantürk, S., Liu, R., Lapointe-Gagné, O., Létourneau, V., Wolf, G., Beaini, D., Rampášek, L. (2024). Graph Positional and Structural Encoder. Proceedings of the 41st ICML 2024, PMLR 235:5533-5566.

[2] Choromanski, K., Likhosherstov, V., Dohan, D., Song, X., Gane, A., Sarlós, T., Hawkins, P., Davis, J., Mohiuddin, A., Kaiser, L., Belanger, D., Colwell, L.J., & Weller, A. (2020). Rethinking Attention with Performers. ArXiv, abs/2009.14794.

[3] Shirzad, H., Velingker, A., Venkatachalam, B., Sutherland, D.J., and Sinop, A.K. (2023). EXPHORMER: sparse transformers for graphs. In Proceedings of the 40th International Conference on Machine Learning (ICML'23), Vol. 202. JMLR.org, Article 1310, 31613–31632.

[4] Wu, Q., Zhao, W., Li, Z., Wipf, D.P., & Yan, J. (2023). NodeFormer: A Scalable Graph Structure Learning Transformer for Node Classification. ArXiv, abs/2306.08385.

---

> ### Author Response · Authors · 2025-11-19
> **Official Comment (Part 1/3)**
>
> Thank you for the review, detailed feedback and suggestions. We address your concerns below. Please also see the general response where we share our view on the positioning of this work in the GFM field.
>
> > **W1**. I agree that the ability to leverage TFMs for graph tasks is a valuable contribution in itself, but I don’t think the paper’s contributions go much beyond that, resulting in a paper limited in scope and largely relying on the success of its implementation. In relation with this, I think the paper somewhat oversells its contributions -- while I understand the reasoning to associate the resulting framework with graph foundation models (GFM), I think it’s bit of a stretch to argue that the resulting model is a GFM in the conventional sense. Applying the proposed framework to any graph requires pre-computing structure-based features, which comes with a non-trivial computational cost per graph; avoiding such hand-crafted feature engineering is one of the main driving forces of graph representation learning, and in a related manner GFMs in the first place. The hand-crafted features alone mean the learned embeddings are not transferable by themselves without computing these features for unseen graphs first.
>
> We are not willing to oversell our work and would welcome alternative versions on how to call G2T-FM. However, as for now, we believe that reasons to call resulting models GFMs outweigh reasons not to do that. First, and most importantly, we believe that the GFM field is exactly the field where our work may have the greatest impact, and we want the GFM community to know about our work. Second, our model can be applied in an in-context learning setting, which is an inherent property of foundation models, not task-specific ones. Third, GFMs represent an active area of research, and we believe the community currently lacks a strict and clear definition of GFM. For example, regarding your concern on precomputing hand-crafted structure-based features, we would like to note that there are other works like OpenGraph and AnyGraph that use (truncated) SVD of the adjacency matrix (or its variations) and yet call their models GFMs. Thus, while the lack of graph-specific pretraining in our method can be a reason not to assign it to GFMs, we believe that reasons to call resulting models GFMs outweigh this one. Following your comment and in order to clearly outline the scope of our work, we have extended the Limitations section of our paper, please refer to Appendix A.

---

> > ### Author Response · Authors · 2025-11-19
> > **Official Comment (Part 2/3)**
> >
> > > **W2** and **Q1**. Weak benchmarks: The current experimental section needs to be significantly strengthened to make a more convincing argument towards the merits of G2T-FMs. The crux of the paper is that using tools like NFA, structural features and heuristic (LapPE)/learnable (PEARL) positional encodings allow us to apply TFMs in graph data. Note that most if not all of these tools can be directly applied to not just G2T-FMs but also both GNN and GFN benchmarks compared against. Measuring G2T-FMs against benchmark methods that do not also use structural features or positional encodings results in unfair evaluation — whether G2T-FMs can outperform other architectures when the same structural information is provided to all will provide a much healthier signal on the usefulness of TFMs on graph data. I suggest several evaluation settings to the authors in the Questions section.
> >
> > Thank you for valuable and thoughtful suggestions, we really appreciate it!
> >
> > In fact, we already have some evaluation results in our paper similar to the proposed ones. Due to space limitations, it was taken out in Appendix C instead of the main text. Specifically, Table 5 provides comparison of G2T-FM with baselines (GNNs and LightGBM) enhanced with the same components as G2T-FM, i.e., NFA, node degree, PageRank, LapPE and PEARL. For LightGBM, PEARL is fixed to the same random initialization used in G2T-FM in ICL regime. Since task-specific baselines receive exactly the same features as G2T-FM, we believe this experiment shows that performance gains come not only from these feature-engineered components, but also from the utilized TFM. If these experiments are not convincing enough, please let us know. Also, please note that we have added the G2T-LimiX results to Table 5 in the new revision, so we recommend using the updated version.
> >
> > Regarding global graph transformers, [1] shows that well-tuned classic GNNs, i.e., provided with common architectural enhancements like LayerNorm or residual connections and hyperparameter tuning, outperform global graph transformers on node classifications tasks. Since we use both architectural improvements and hyperparameter tuning, we believe comparing with global graph transformers is redundant.
> >
> > Regarding prior GFMs with the same feature components, one can clearly see that they work **significantly** worse than GNNs, and we already have GNNs evaluated with these graph-based features.
> >
> > Yet, if you think that some of these experiments are critical and will bring additional insights, please let us know and we will try to provide the results during the discussion period.
> >
> > In summary, as we note in our general response, we believe that outperforming well-tuned classic GNNs already poses a significant milestone that was not achieved by prior GFMs for the general case of graphs with arbitrary feature/target spaces and realistic data splits.

---

> > > ### Author Response · Authors · 2025-11-19
> > > **Official Comment (Part 3/3)**
> > >
> > > > **W3** and **Q2**. In relation to the previous point, the authors don’t address how their G2T-FMs compare with the other benchmarks, in particular the GFMs, in terms of efficiency. What are the parameter counts for the optimal models? How long does pre-training and/or fine-tuning take for each? How fast is downstream inference? Answering these questions will allow evaluating the merits of the proposed model better, but the information simply isn’t there. Similarly, the cost of graph pre-processing is not discussed, which is crucial considering they compare against benchmarks that do not have this pre-processing overhead.
> > >
> > > In this work, we aimed to answer the general question on whether it is possible to design a GFM that can compete (in terms of predictive performance) with well-tuned GNNs on arbitrary tasks in realistic scenarios. Our approach allows us to give a positive answer to this question, and we leave scaling to larger graphs for future research (please see our general response for a more detailed discussion).
> > >
> > > We do not compare G2T-FM with other GFMs in terms of efficiency, since prior GFMs have **significantly** worse downstream performance compared to both GNNs and G2T-FM. However, to address your comment and for the completeness of our study, we are currently benchmarking the efficiency of our approach and GNNs, and will provide the results when the experiments are finished.
> > >
> > > For now, below we analyze the time/space complexity of our method. We use the following notation: $N$ stands for the number of nodes, $E$ for the number of edges and $F$ for the number of features. For convenience, we also assume that $E + 1 \geq N$, which always holds for connected graphs. For conciseness, we assume all other hyperparameters to be fixed and do not include them in $O$-notation.
> > >
> > > The graph preprocessing step consists of computing NFA, LapPE, degree, PageRank and PEARL embeddings.
> > > - NFA has $O(EF)$ time complexity and $O(NF)$ space complexity.
> > > - Computing a fixed number of LapPE features requires $O(E)$ time and space complexity.
> > > - Computing degree has $O(E)$ time and $O(N)$ memory complexity.
> > > - Computing PageRank requires $O(E)$ time and $O(N)$ space complexity for each iteration, and we use at most 100 iterations.
> > > - PEARL is essentially a GNN applied several times, so it has $O(E)$ time complexity and $O(N)$ space complexity.
> > >
> > > Overall, the preprocessing has $O(EF)$ time complexity and $O(NF+E)$ space complexity.
> > >
> > > Since both TabPFNv2 and LimiX have dual attention (i.e., interleaving attention operations over sample dimension and over feature dimension), they have $O(N^2 F + N F^2)$ time complexity and $O(NF)$ memory complexity.
> > >
> > > ***
> > >
> > > [1] Classic GNNs are Strong Baselines: Reassessing GNNs for Node Classification, NeurIPS 2024

---

> > > > ### Author Response · Authors · 2025-11-26
> > > >
> > > > Dear Reviewer uY6A,
> > > >
> > > > We have recently completed our experiments on benchmarking time and memory efficiency of considered methods, please refer to [this comment](https://openreview.net/forum?id=8tW32RrecK&noteId=SZlXIhNt0D) for the results. In short, G2T-FM potential problems with efficiency seem to be not so evident as one may think. Please let us know if your concerns were addressed by our original response and these [additional results](https://openreview.net/forum?id=8tW32RrecK&noteId=SZlXIhNt0D). We would be happy to discuss further if there are any remaining questions. Following our [general response](https://openreview.net/forum?id=8tW32RrecK&noteId=itqnPkUCKC), we would also be happy to discuss positioning of this work in the GFM field and future potential of graph foundation models in general.
> > > >
> > > > Best regards,
> > > >
> > > > Authors

---

### Official Review · Reviewer_hjSn · 2025-11-01

**Soundness:** 2
**Presentation:** 2
**Contribution:** 1
**Rating:** 2
**Confidence:** 4

**Summary:**

This paper proposes G2T-FM, a framework that transforms tabular foundation models (TFMs) into graph foundation models (GFMs) by incorporating neighborhood aggregation and structural embeddings. It achieves strong in-context and fine-tuned performance, surpassing existing GFMs and even well-tuned GNNs on various graph tasks.

**Strengths:**

1. The paper first studies the potential of tabular foundation models in graph-related applications.
2. The proposed approach is common and, in general, sound in graph-related applications.

**Weaknesses:**

1. The novelty of this work is limited. Although it claims to be the first attempt at turning TFMs into GFMs, the proposed method is straightforward: only adding structure features to node features and reusing existing TFMs. This approach, though effective, is well-established and there is no surprise that including such side information will lead to improvements.
2. The efficiency/cost of the proposed method is not discussed. Specifically, it requires additional preprocessing for computing the complementary features, which can be costly for large graphs. It also didn't compare the test-time efficiency with existing methods, neither in the ICL setting nor in the fine-tuning setting. Given the size of TFMs, even fine-tuning them on test datasets could be costly.
3.  Several details about implementation are missing, e.g., the order of feature aggregation, the steps of fine-tuning.
4. Limited gains: the authors mentioned that the observed performance gains of their method might come from the inclusion of side information that was never used in baseline models. Table 5 shows that, with enhanced features, simple GNNs perform reasonably well and the gains of the proposed method become less significant. Given the potentially high inference cost, the applicability of the proposed method is challenged.

**Questions:**

What is the time/space complexity of the proposed method? How to extend it to larger graphs?

---

> ### Author Response · Authors · 2025-11-19
> **Official Comment (Part 1/2)**
>
> Thank you for your review. We address your questions and comments below. Please also see the general response where we share our view on the positioning of this work in the GFM field.
>
> > **W1**. The novelty of this work is limited. Although it claims to be the first attempt at turning TFMs into GFMs, the proposed method is straightforward: only adding structure features to node features and reusing existing TFMs. This approach, though effective, is well-established and there is no surprise that including such side information will lead to improvements.
>
> Although supplying graph-agnostic models (e.g., MLPs) with some information about graph structure is already an explored technique, our work is, to the best of our knowledge, the first one that specifically applies TFMs to graph data. We believe this idea is both novel and important, since it opens a new direction for building GFMs. Please see our general response for a more detailed discussion.
>
> > **W2**. The efficiency/cost of the proposed method is not discussed. Specifically, it requires additional preprocessing for computing the complementary features, which can be costly for large graphs. It also didn't compare the test-time efficiency with existing methods, neither in the ICL setting nor in the fine-tuning setting. Given the size of TFMs, even fine-tuning them on test datasets could be costly.
>
> > **Q1**. What is the time/space complexity of the proposed method? How to extend it to larger graphs?
>
> In this work, we aimed to answer the general question on whether it is possible to design a GFM that can compete in terms of predictive performance with well-tuned GNNs on arbitrary tasks in realistic scenarios. Our approach allows us to give a positive answer to this question, and we leave scaling to larger graphs for future research (please see our general response for a more detailed discussion).
>
> To address your comment and for the completeness of our study, we are currently benchmarking the efficiency of our approach and will provide the results when the experiments are finished.
>
> For now, below we analyze the time/space complexity of our method. We use the following notation: $N$ stands for the number of nodes, $E$ for the number of edges and $F$ for the number of features. For convenience, we also assume that $E + 1 \geq N$, which always holds for connected graphs. For conciseness, we assume all other hyperparameters to be fixed and do not include them in $O$-notation.
>
> The graph preprocessing step consists of computing NFA, LapPE, degree, PageRank and PEARL embeddings.
> - NFA has $O(EF)$ time complexity and $O(NF)$ space complexity.
> - Computing a fixed number of LapPE features requires $O(E)$ time and space complexity.
> - Computing degree has $O(E)$ time and $O(N)$ memory complexity.
> - Computing PageRank requires $O(E)$ time and $O(N)$ space complexity for each iteration, and we use at most 100 iterations.
> - PEARL is essentially a GNN applied several times, so it has $O(E)$ time complexity and $O(N)$ space complexity.
>
> Overall, the preprocessing has $O(EF)$ time complexity and $O(NF+E)$ space complexity.
>
> Since both TabPFNv2 and LimiX have dual attention (i.e., interleaving attention operations over sample dimension and over feature dimension), they have $O(N^2 F + N F^2)$ time complexity and $O(NF)$ memory complexity.
>
> > **W3**. Several details about implementation are missing, e.g., the order of feature aggregation, the steps of fine-tuning.
>
> We use 1-hop NFA aggregations, as originally proposed in [1]. In the updated paper, we clarified that by ‘neighbors’ we mean ‘1-hop neighbors’ (line 258).
>
> For TFM finetuning, as written in line 738, we use the procedure proposed in [2]. Specifically, we do not set a maximal number of finetuning steps, but instead use early stopping — we compute the performance on the validation set every ten steps and stop finetuning after 16 non-improving evaluations. These details can also be found in the code we release.

---

> > ### Author Response · Authors · 2025-11-19
> > **Official Comment (Part 2/2)**
> >
> > > **W4**. Limited gains: the authors mentioned that the observed performance gains of their method might come from the inclusion of side information that was never used in baseline models. Table 5 shows that, with enhanced features, simple GNNs perform reasonably well and the gains of the proposed method become less significant. Given the potentially high inference cost, the applicability of the proposed method is challenged.
> >
> > Please note that since Table 5 supplements our ablation analysis, we did not include the results of G2T-LimiX there. When comparing GNNs with enhanced features against G2T-LimiX, the gains are substantial. For your convenience, we provide a table with these results below, and we also updated Table 5 in the paper for clarity.
> >
> > |                 | tolokers-2   | city-reviews | artnet-exp   | hm-prices    | avazu-ctr    | city-roads-M | twitch-views | artnet-views | AR   |
> > | :-------------- | :----------- | :----------- | :----------- | :----------- | :----------- | :----------- | :----------- | :----------- | :--- |
> > | GCN (M)         | 58.71 ± 0.45 | 77.07 ± 0.27 | 43.44 ± 0.32 | 70.73 ± 0.26 | 31.10 ± 0.22 | 57.91 ± 0.22 | 77.11 ± 0.09 | 56.14 ± 0.24 | 4.62 |
> > | GraphSAGE (M)   | 59.59 ± 0.51 | 77.95 ± 0.09 | 44.31 ± 0.53 | 70.50 ± 0.47 | 31.51 ± 0.41 | 59.66 ± 0.09 | 75.93 ± 0.19 | 55.39 ± 0.32 | 4.12 |
> > | GAT (M)         | 57.76 ± 0.70 | 77.47 ± 0.14 | 44.36 ± 0.50 | 72.46 ± 0.49 | 31.97 ± 0.23 | 59.57 ± 0.43 | 77.04 ± 0.08 | 56.51 ± 0.35 | 3.62 |
> > | GT (M)          | 58.79 ± 0.76 | 76.43 ± 0.10 | 43.03 ± 0.60 | 71.84 ± 0.64 | 29.86 ± 0.67 | 59.85 ± 0.41 | 76.15 ± 0.11 | 56.39 ± 0.31 | 4.50 |
> > | G2T-LimiX (ICL) | 61.48 ± 0.30 | 77.72 ± 0.54 | 48.43 ± 0.18 | 74.96 ± 0.06 | 32.39 ± 0.14 | 64.53 ± 0.07 | 71.08 ± 0.07 | 60.95 ± 0.10 | 2.50 |
> > | G2T-Limix (FT)  | 61.17 ± 0.49 | 80.13 ± 0.05 | 49.88 ± 0.13 | 76.32 ± 0.17 | 33.94 ± 0.34 | 65.87 ± 0.10 | 73.16 ± 0.40 | 62.12 ± 0.10 | 1.62 |
> >
> > Please let us know if you have any additional questions or comments.
> >
> > ***
> >
> > [1] GraphLand: Evaluating Graph Machine Learning Models on Diverse Industrial Data, NeurIPS 2025
> >
> > [2] On Finetuning Tabular Foundation Models, arXiv preprint 2025

---

> > > ### Author Response · Authors · 2025-11-26
> > >
> > > Dear Reviewer hjSn,
> > >
> > > We have recently completed our experiments on benchmarking time and memory efficiency of considered methods, please refer to [this comment](https://openreview.net/forum?id=8tW32RrecK&noteId=SZlXIhNt0D) for the results. In short, G2T-FM potential problems with efficiency seem to be not so evident as one may think. Please let us know if your concerns were addressed by our original response and these [additional results](https://openreview.net/forum?id=8tW32RrecK&noteId=SZlXIhNt0D). We would be happy to discuss further if there are any remaining questions. Following our [general response](https://openreview.net/forum?id=8tW32RrecK&noteId=itqnPkUCKC), we would also be happy to discuss positioning of this work in the GFM field and future potential of graph foundation models in general.
> > >
> > > Best regards,
> > >
> > > Authors

---

### Author Response · Authors · 2025-11-19
**General response to all reviews**

Dear Reviewers,

We thank you for your valuable feedback. This is a general response to all reviews. We addressed the specific comments and questions in individual responses below.

Many of you expressed concerns regarding efficiency, scalability to larger graphs, and applicability to link- and graph-level tasks. While these comments are valid, we believe that **the main contribution of our work was underestimated**, and we want to elaborate on it.

For a long time, **we believed that graph foundation models were impossible to create** due to several fundamental challenges. Even when focusing only on node-level prediction tasks, graphs are found in extremely diverse domains ranging from social communications and web graphs to transport networks. This makes it difficult to imagine a single model that can handle all of these tasks and achieve knowledge transfer between different datasets. Furthermore, unlike CV or NLP, in the graph ML field there remains an insufficient amount of high-quality, large-scale, diverse pretraining graph datasets, which makes it difficult to cover the vast landscape of potential applications.

We believe that **previous works have not fully overcome these challenges**. Most progress has been limited to text-attributed graphs, where tasks can be unified via text encoders, or to few-shot scenarios, which are less practical for many real-world applications. As far as we know, **existing GFMs perform significantly worse than well-tuned GNNs** trained from scratch in the general case of node-level tasks with non-fewshot splits and graphs from arbitrary domains and arbitrary features. We would welcome evidence to the contrary, but our evaluation results support this statement and align with our intuition. For example, many prior GFMs are pretrained on at most several dozen graphs, which is orders of magnitude smaller than the amount of data used for pretraining foundation models in other domains. **The similar position was expressed by other researchers**, as outlined in [1], which points to the lack of truly generalizable GFMs.

By contrast, our proposed **G2T-FM is the first GFM that can reliably compete with and often significantly outperform GNNs trained from scratch on node-level prediction tasks**. This represents a significant advancement over the existing open-source GFMs and, in our opinion, constitutes an important milestone for the field. While G2T-FM is possibly not yet ready for deployment in production settings due to scalability issues, **it provides a valuable proof of concept and a strong baseline for future research in GFMs**.

We would also like to note that the considered TabPFNv2 and LimiX models are examples of Prior-data Fitted Networks (PFNs, [2]), which is a relatively young approach and the subject of active research now. Scalability to large datasets is a general problem for this approach (rather than a specific issue with our method), but progress in solving this problem will likely be made in the future, and all solutions will be directly applicable to our method. We would like to emphasize that **our work is the first to apply PFNs to graph machine learning tasks and we believe it is a significant contribution**.

In summary, **the key contribution of our work is that it shows that GFMs are feasible**. While considerable work remains, such as scaling to larger graphs and addressing link- and graph-level tasks, our results show a clear and promising direction forward. We hope that our work will support the future development of GFMs and will inspire further research in this area.

Best regards,

Authors

***

[1] Position: Graph Learning Will Lose Relevance Due To Poor Benchmarks, ICML 2025

[2] Transformers Can Do Bayesian Inference, ICLR 2022

---

> ### Author Response · Authors · 2025-11-26
> **Benchmarking time and memory efficiency**
>
> Dear Reviewers,
>
> As we mention in our general response, we acknowledge potential limitations of G2T-FM in terms of efficiency and scalability, yet we believe the work to be valuable for the GFM field. However, for completeness of our study, below we provide the results of benchmarking G2T-FM and GCN in terms of time and memory efficiency.
>
> For each dataset, all methods were evaluated in exactly the same environment on the same machine, making the comparison fair. For all methods, we use the optimal hyperparameters chosen during our main experiments, and only make one evaluation with a single random seed.
>
> The table below contains peak memory consumption measured in GiB. One can clearly see that the memory consumption of G2T-LimiX is noticeably higher than those of GCN.
>
> |           | tolokers-2 | city-reviews | artnet-exp | hm-prices | avazu-ctr | city-roads-M | twitch-views | artnet-views |
> | :-------- | :--------- | :----------- | :--------- | :-------- | :-------- | :----------- | :----------- | :----------- |
> | GCN       |       0.56 | 5.85         |        2.1 |      2.53 | 3.69      |         2.17 | 6.59         |         1.93 |
> | G2T-LimiX |        3.5 | 49.48        |      35.61 |      46.7 | 36.83     |        15.21 | 40.17        |        20.02 |
>
> We also measure time (in seconds) required to preprocess a dataset (“simple” means that we only load it from disk and transform to the desired format, and in “G2T-FM” preprocessing we also compute LapPE, node degree, PageRank and NFA used in G2T-FM), to train (or finetune) a model, and to infer a model on a dataset.
>
> |                       | tolokers-2 | city-reviews | artnet-exp | hm-prices | avazu-ctr | city-roads-M | twitch-views | artnet-views |
> | :-------------------- | :--------- | :----------- | :--------- | :-------- | :-------- | :----------- | :----------- | :----------- |
> | Preprocessing: simple |       0.45 | 2.15         |       1.46 |     11.16 | 14.93     |         0.35 | 7.17         |         1.08 |
> | Preprocessing: G2T-FM |       1.16 | 30.47        |       5.14 |     27.88 | 54.92     |        42.49 | 14.56        |         3.27 |
> | Train: GCN            |      25.77 | 85.66        |      41.58 |      77.9 | 32.90     |        22.71 | 327.46       |        19.53 |
> | Finetune: G2T-LimiX   |     139.52 | 1688.98      |     832.07 |    1123.4 | 1013.35   |        225.9 | 1680.59      |       457.12 |
> | Inference: G2T-LimiX  |       0.87 | 27.27        |      16.39 |     20.77 | 15.14     |         3.58 | 21.32        |         5.54 |
>
>
> From the table above, one can see that neither preprocessing nor inference of G2T-LimiX are significantly longer than the training of a GNN. Thus, G2T-FM seems to not have evident efficiency problems when evaluated in the ICL regime. Finetuning of G2T-FM is, however, noticeably more time-consuming.
>
> As a final remark, let us note that the efficiency of both G2T-FM and GNN was not the primary focus of our work. Thus, the results above might not fully reflect the optimal efficiency of these models. However, we believe that they reflect the overall picture.
>
> Best regards,
>
> Authors

---

> > ### Author Response · Authors · 2025-11-26
> > **Experiments with multi-hop aggregation**
> >
> > Dear Reviewers,
> >
> > We have conducted additional experiments with multi-hop NFA, as requested by some reviewers. Specifically, we evaluated G2T-LimiX (ICL), in which we replaced 1-hop NFA with concatenation of (i) 1-hop and 2-hop NFA; and (ii) 1-hop, 2-hop and 3-hop NFA. We provide the results of the evaluation below.
> >
> > |                           | pubmed       | facebook     | amazon-ratings | questions    | wiki-cs      | tolokers-2   | city-reviews | artnet-exp   | hm-prices    | avazu-ctr    | city-roads-M | twitch-views | artnet-views |
> > | :------------------------ | :----------- | :----------- | :------------- | :----------- | :----------- | :----------- | :----------- | :----------- | :----------- | :----------- | :----------- | :----------- | :----------- |
> > | G2T-LimiX (ICL)           | 88.96 ± 0.18 | 91.29 ± 0.14 | 44.10 ± 0.16   | 15.31 ± 0.77 | 79.99 ± 0.28 | 61.48 ± 0.30 | 77.72 ± 0.54 | 48.43 ± 0.18 | 74.96 ± 0.06 | 32.39 ± 0.14 | 64.53 ± 0.07 | 71.08 ± 0.07 | 60.95 ± 0.10 |
> > | G2T-LimiX 2-hop NFA (ICL) | 87.92 ± 0.24 | 91.87 ± 0.12 | 44.39 ± 0.24   | 18.38 ± 1.32 | 79.16 ± 0.21 | 61.13 ± 0.12 | 76.09 ± 1.37 | 36.78 ± 5.57 | 75.09 ± 0.06 | 32.41 ± 0.18 | 64.98 ± 0.11 | 71.62 ± 0.07 | 62.11 ± 0.05 |
> > | G2T-LimiX 3-hop NFA (ICL) | 87.19 ± 0.21 | 91.85 ± 0.08 | 44.38 ± 0.24   | 17.83 ± 0.81 | 78.68 ± 0.29 | 60.70 ± 0.18 | 75.60 ± 1.03 | 38.81 ± 4.79 | 75.24 ± 0.02 | 32.04 ± 0.23 | 65.05 ± 0.08 | 72.04 ± 0.05 | 62.26 ± 0.02 |
> >
> > We see that increasing the number of hops in NFA can improve the performance on some datasets (e.g., `questions` or `artnet-views`). However, it may also result in degraded performance on others (e.g., `city-reviews` or even a significant drop on `artnet-exp`). Since additional hops do not consistently yield better results and also increase the number of features, thus reducing time and memory efficiency, we consider one-hop NFA to be the preferred default choice for G2T-FM. Nonetheless, the use of multi-hop NFA within the G2T-FM framework remains possible and may, in some cases, achieve better performance.
> >
> > Best regards,
> >
> > Authors

---

### Author Response · Authors · 2025-12-03
**Rebuttal summary**

Dear Reviewers and Area Chair,

We would like to thank the reviewers for their valuable feedback and effort in reviewing our submission. Unfortunately, the current rules do not allow the reviewers to continue the discussion. However, we believe that we have addressed the reviewers' concerns. Below, we summarize the reviews and our responses.

First, we would like to point to our [general response](https://openreview.net/forum?id=8tW32RrecK&noteId=itqnPkUCKC), where we share our view on the GFM field and positioning of our work within it. In short, for a long time we believed that GFMs are infeasible in realistic setups, but the success of recent TFMs based on the PFN framework has changed our minds. In this work, we show that their success can be transferred to graphs, and that GFMs are feasible even in the challenging setup we consider. More broadly, we believe that PFNs offer a promising path towards building truly generalizable GFMs, and our work makes the first step towards adopting PFNs for graph data.

Now, let us summarize strengths and weaknesses mentioned by the reviewers.

**Strengths:**

- **Novelty**
	- `EACW`: "It is a novel approach to apply tabular foundation model to the graph domain"
	- `uY6A`: "The core idea of leveraging well-established tools from graph learning literature to adapt graph tasks to TFMs, which is conceptually simple but effective"
	- `Cmbg`: "The paper introduces a clear and compelling analogy between tabular data and heterogeneous graph features, enabling the transfer of advances from tabular foundation models (TFMs) into the graph machine learning domain"
- **Empirical results**
	- `uY6A`: "The results clearly back the authors’ claims; ICL results on datasets with text-based features are particularly promising"
	- `Cmbg`: "On multiple datasets, the proposed approach achieves competitive or superior results compared to well-tuned traditional GNN baselines and existing GFM implementations, despite its simplicity"
- **Presentation**
	- `uY6A`: "The paper is well-written; the contributions are clearly stated and the logical flow is easy to follow"
	- `Cmbg`: "It provides a thorough and insightful discussion of the limitations of prior graph foundation models, highlighting gaps in generalization and feature handling"


**Weaknesses:**

- **Scalability and efficiency:** G2T-FM is not a production-ready solution, but rather a proof of concept. We therefore leave these issues for future research. However, we would also like to note that the preprocessing overhead is not as high as one might think, please see our [benchmarking results](https://openreview.net/forum?id=8tW32RrecK&noteId=SZlXIhNt0D) for details. Importantly, progress in scalability of TFMs can be directly transferred to G2T-FM.
- **GFM claim:** Several reviewers raised concerns about whether G2T-FMs can be called GFMs since, for example, resulting models cannot handle link- or graph-level tasks. However, we believe that the community currently lacks a strict and clear definition of a GFM, and there are other GFMs (e.g., TS-GNN and GCOPE) that do not handle link- or graph-level tasks yet call their models GFMs. Moreover, G2T-FMs can be applied in an in-context learning setting, which is an inherent property of foundation models rather than task-specific ones. Most importantly, we believe that the GFM field is exactly where our work will have the greatest impact, and we want the GFM community to know about it. This being said, we are not willing to oversell our work and would welcome alternative versions on how to call G2T-FM.
- **Limited novelty and technical depth:** We believe that the simplicity of G2T-FM is actually its strength, as it can be straightforwardly adapted to any existing or future TFM. Moreover, as we show in [the comment “Experiments with multi-hop aggregation”](https://openreview.net/forum?id=8tW32RrecK&noteId=zAP4Wwuaqr), providing G2T-FM with multi-hop information does not consistently improve performance.
- **Comparison with stronger baselines:** In our study, we use well-tuned GNNs with optimized hyperparameters and architectural enhancements (like residual connections and layer normalization). As shown in [1], such GNNs already serve as strong baselines. Moreover, in our ablation study, we also evaluate GNNs with exactly the same features as G2T-FM (i.e., enhanced with LapPE, PageRank, etc.). Table 5 clearly shows that G2T-FM (in particular, G2T-LimiX) significantly outperforms even these GNNs with enhanced features.

Overall, we believe that the reviewers' concerns while being valid do not override the importance of our main contribution, which is definitely valuable for the GNN field given the striking empirical results.

Best regards,

Authors

***

[1] Classic GNNs are Strong Baselines: Reassessing GNNs for Node Classification, NeurIPS 2024

---

### Meta-Review · Area_Chair_3RgD · 2026-01-08

**Summary:**

Reviewers note the following key issues:

- Limited novelty. While utilizing TFMs for GFMs is new, the solution is straightforward and a direct application/reusing of existing TFMs.
- Lack of efficiency discussions and comparisons
- The claim of GFM might be too strong due to lack of more diverse tasks and cross graph structure learning.

**Reviewer Concerns:**

Rebuttals on novelty and GFM claim is not convincing.

Some complexity analysis is offered, but no systematic empirical evaluation.

**Reviewer Scores:**

I expect minimal changes since most of the concerns remain.

---

### Decision · Program_Chairs · 2026-01-26

Reject